# The Midbrain Preisthmus: A Poorly Known Effect of the Isthmic Organizer

**DOI:** 10.3390/ijms24119769

**Published:** 2023-06-05

**Authors:** Luis Puelles, Matias Hidalgo-Sánchez

**Affiliations:** 1Department of Human Anatomy, School of Medicine, Murcia Institute of Biomedical Research, University of Murcia, El Palmar, 30120 Murcia, Spain; 2Department of Cell Biology, Faculty of Sciences, University of Extremadura, 06006 Badajoz, Spain; mhidalgo@unex.es

**Keywords:** preisthmus, isthmus, midbrain, isthmic organizer, patterning, FGF8, WNT1, Otx2

## Abstract

This essay reexamines molecular evidence supporting the existence of the ‘preisthmus’, a caudal midbrain domain present in vertebrates (studied here in the mouse). It is thought to derive from the embryonic m2 mesomere and appears intercalated between the isthmus (caudally) and the inferior colliculus (rostrally). Among a substantial list of gene expression mappings examined from the Allen Developing and Adult Brain Atlases, a number of quite consistent selective positive markers, plus some neatly negative markers, were followed across embryonic stages E11.5, E13.5, E15.5, E18.5, and several postnatal stages up to the adult brain. Both alar and basal subdomains of this transverse territory were explored and illustrated. It is argued that the peculiar molecular and structural profile of the preisthmus is due to its position as rostrally adjacent to the isthmic organizer, where high levels of both FGF8 and WNT1 morphogens must exist at early embryonic stages. Isthmic patterning of the midbrain is discussed in this context. Studies of the effects of the isthmic morphogens usually do not attend to the largely unknown preisthmic complex. The adult alar derivatives of the preisthmus were confirmed to comprise a specific preisthmic sector of the periaqueductal gray, an intermediate stratum represented by the classic cuneiform nucleus, and a superficial stratum containing the subbrachial nucleus. The basal derivatives, occupying a narrow retrorubral domain intercalated between the oculomotor and trochlear motor nuclei, include dopaminergic and serotonergic neurons, as well as a variety of peptidergic neuron types.

## 1. Introduction

### The Isthmic Organizer and the Preisthmus

The isthmic organizer was the first ‘secondary organizer’ discovered in the developing neural tube [1,2]. The *secondary organizers* are linear neuroepithelial areas that produce diffusible morphogen signals acting on regional patterning and differential specification of neighboring progenitors. The isthmic organizer is in a way complex, because it presents two different but adjacent morphogen sources—releasing FGF8 and WNT1, respectively—on its isthmic and midbrain sides (Figure 1).

After diffusion, these molecules show graded fields of concentration across a spatial range of the surrounding neuroepithelium; the isthmic organizer forms at the transverse isthmo-mesencephalic boundary and exerts anteroposterior effects on the whole midbrain and rostral isthmo-cerebellar or prepontine hindbrain (Figure 1). The reaction-competent neuroepithelium is composed of progenitor cells that can assess and react to the amount of diffused FGF8 morphogen detected. Morphogen molecules vary in their diffusion in the local environment (WNT1 is thought to have a low diffusibility because it rapidly attaches to the intercellular matrix, while FGF8 diffuses significantly, apparently reaching the interneuromeric limit r1/r2 caudalwards and advancing well into the midbrain rostralwards). The spread of such diffusing morphogens may be curtailed by the presence of inhibitors [3]. However, an inhibitor surround has not been demonstrated yet for the isthmic organizer.

The gradient of isthmic morphogen provides *non-instructive positional information* to the progenitors by means of the diverse levels in morphogen concentration that are established (light blue thresholds in Figure 1). The competent cells respond to this variable by choosing among two or more differentiation options (activation potencies) provided by their partly prespecified genome (i.e., midbrain cells react differently than hindbrain ones). In the genome there supposedly are various cis-regulatory enhancers sensitive to larger or smaller amounts of the morphogen signal; once an enhancer is activated, it triggers a particular cascade of gene activations entailing particular differentiation effects. The overall result is that the affected neuroepithelial field results subdivide developmentally into as many subfields as differential reactions occur. As a consequence, each subfield will develop a distinct tissue fate corresponding to the particular gene cascade activated.

The isthmic organizer, characterized by its expression of the fibroblast growth factor 8 gene (*Fgf8*; [4]), forms next to the transverse constriction that separates the midbrain vesicle (caudalmost part of the forebrain tagma) from the hindbrain tagma [4]. The whole alar plate of the isthmic rhombomere (r0) initially expresses *Fgf8* and thus represents a source of FGF8 protein [5]. However, the areal extent of the FGF8 source cells subsequently changes. Possibly by action of an unknown inhibitory surround effect [3], the organizer gets reduced in its caudal areal extent and is progressively restricted closer and closer to the rostrally limiting isthmo-mesencephalic boundary over time [6,7]. The isthmus or rhombomere 0 represents the rostralmost prepontine hindbrain neuromeric unit [6,8]. It was shown that transplanting pieces of the isthmic alar neuroepithelium ectopically anywhere in the competent diencephalo-mesencephalo-rhombencephalic domain (from the zona limitans intrathalamica to the rhombospinal boundary) generates ectopic copies of the cerebellum [2,9,10,11,12].

Ulterior studies have elaborated on the hypothesis that the observed isthmic induction effects were due to release of the FGF8 morphogen. Collagen microbeads impregnated with a solution of FGF8 protein were found to also induce cerebellar primordia at the competent sites cited above ([4]; review in [12]).

Accounts of isthmic organizer effects usually concentrate on the presence of cerebellar primordia. Less often, midbrain auditory elements generated experimentally are examined. The isthmic region itself, which develops some structural complexity (various *isthmic nuclei* derive from a primary ‘isthmic cell plate’ [13,14]), is not studied as a rule, though it is the area where the highest morphogen concentrations occur on the hindbrain side. Results affecting the nuclei of cranial nerves 3 and 4 located in the basal plate (sometimes duplicated) are rarely commented upon.

There is, moreover, a Cinderella-like subregion of the caudal midbrain—called *preisthmus* [15]—that is completely disregarded in such experiments (PrIsth in Figure 1). It falls within the midbrain range of action of the isthmic organizer, caudally to the auditory midbrain (IC in Figure 1) and close to the WNT1-releasing caudal ring. Since it also lies very close to the isthmic FGF8 source, the preisthmus must be one of the differential fates patterned by concentrations of FGF8 rising beyond those that produce a cerebellum or an auditory caudal midbrain.

This caudal midbrain area was modernly redefined topographically and molecularly in the chick [15] as a conceptually ‘new’ *preisthmic* alar midbrain region; it was indeed characterized as being strictly mesencephalic in molecular profile by its expression of the midbrain-selective marker *Otx2*. The avian preisthmus further expresses *Pax2*, which is absent in the rest of the midbrain. It was found later that the preisthmus also selectively expresses *Six3* and lacks *Tcf7l2* transcripts [16]. The new anatomic name ‘preisthmus’ refers to its unique topography.

The existence of a thin caudal mesomere 2 at this locus had been proposed before after comparative embryonic studies by [13,17,18]. However, these authors added the questionable interpretation that m2 was an atrophic neuromere that *did not produce any neuronal derivatives*. They conceived it rather as a transverse neuron-free gap between the midbrain and the hindbrain. Since neuromeres are held to be serial transversal *histogenetic units* of the neural tube, showing proliferative bulging, internal regionalization, and differential areal production of characteristic neuronal populations (e.g., [19,20]), the notion that one of them remained atrophic long caused private skepticism on its neuromeric nature (e.g., [14,19]). The results of Hidalgo-Sánchez et al. [15] first showed that this m2 domain generates alar and basal preisthmic grisea distinct from the auditory midbrain, as expected of a true neuromere.

The anatomy of the adult mouse preisthmus was later covered ([21,22]; see also [5,23,24]). Mammalian preisthmic structures that are homologous to their chicken counterparts according to an identical topology in the brain Bauplan largely correspond with the classic ‘cuneiform nuclear complex’ (CnF in rodent atlases) found in coronal sections dorsally to the isthmic parabigeminal nucleus and under the inferior colliculus. This complex appears divided into superficial (subpial) and intermediate strata. These cover a particular preisthmic portion of deep periaqueductal gray, which lies ventral (topologically caudal) to that corresponding to the inferior colliculus. Due to the lack of neuromeric delimiting concepts, recent brain atlases tend to be imprecise in their employment of the CnF tag, mixing identification of some CnF elements in the correct *preisthmic* place jointly with false CnF elements identified within the *isthmus*. Alternatively, the CnF area may be wrongly extended caudalwards beyond the isthmic course of the trochlear nerve, or a separate precuneiform nucleus arbitrarily defined (e.g., [25]). Puelles et al., 2012 [21] tried to clarify this scenario by suggesting the reservation of the *cuneiform nucleus* name and tag (CnF) for the intermediate stratum population of the true preisthmus. No ‘cuneiform’ entity should accordingly be identified within the isthmus proper. Puelles et al., 2012 [21] also proposed renaming the superficial population of the cuneiform complex as *subbrachial nucleus* (SubB), based on its topography directly under the interstitial nucleus of the brachium of the inferior colliculus (BIC) [26].

The basal preisthmus is less extensive rostrocaudally than the alar part. It forms a narrow wedge domain that can be described as *retrorubral* in topography, lying rostrocaudally between the mesencephalic dorsal and ventral tegmental decussations (and oculomotor nucleus) and the isthmic decussation of the brachium conjunctivum (superior cerebellar peduncle and the trochlear nucleus). The retrorubral narrow space is occupied by dopaminergic neurons, including those of the *interfascicular nucleus* and the *A8 cell group* [21], plus some serotonergic neurons [27] and peptidergic neurons [21].

The preisthmus domain can be explained as forming part of the immediate area of influence of the isthmic organizer in the midbrain. The differentiating m2 domain may be analogous to the neighboring isthmic r0 domain of the hindbrain in its requirement for relatively high concentrations of FGF8 and WNT1 morphogens, acting in this case inside the midbrain. The small anteroposterior dimension and retarded neurogenesis of m2 [20] may be due to singular regulation of proliferation and neurogenesis.

The aim of the present essay is to collect from the Allen Developing and Adult Mouse Brain Atlases novel genoarchitectonic data that illustrate the molecular singularity of the developing preisthmus in the context of the neighboring midbrain and isthmic regions, thus expanding the present concept of the diverse neural fates related to the activity of the isthmic organizer. This point will be re-examined here in light of numerous gene markers.

## 2. Materials and Methods

We centered all present descriptions on material from the Allen Institute for Brain Science Developing Mouse Brain Atlas (developingmouse.brain-map.org, accessed on 30 May 2023; in general, embryonic stages E11.5, E13.5, E15.5, E18.5, and postnatal stages P4, P14, P28, and P56, sometimes amplified to additional intermediate stages for some genes), with occasional reference to the Adult Mouse Brain Atlas (mouse.brain-map.org). Various relevant markers mapped by immunoreaction or in situ hybridization from our own mouse brain collection at embryonic or postnatal stages were also available during this study but were not described specifically.

An Excel sheet was prepared and annotated as Appendix A, listing gene markers examined and their pattern of expression in the preisthmus. Photographic examples of the best Allen preparations were selected and composed into following Figures. These contain examples of sagittal and/or coronal sections of in situ reacted markers at different stages, counterstained with the yellow variant of the Nissl counterstain used at the Allen Institute (note the atlas *coronal* sections intersect the isthmo-mesencephalic transition nearly *horizontally*—i.e., parallel to the length axis-; this is due to its position on the caudal limb of the cephalic flexure).

## 3. Results

### 3.1. Midbrain and Isthmic Gene Patterns in E11.5 Mouse Embryos

When cut in standard coronal sections, the isthmo-mesencephalic region is cut in fact horizontally; the axial dimension goes from the isthmus (bottom) up into the midbrain; in the sagittal sections shown here the ventral region lies to the left and the dorsal region to the right.

The isthmo-mesencephalic boundary (IMB) appears clearly defined by the caudal limit of midbrain *Otx2* expression in the largely undifferentiated neuroepithelium; the incipient mantle layer appears negative (Figure 2a–f). It is clearly observed that this molecular fate boundary does not coincide with the macroscopic isthmic constriction, so that the rostralmost isthmus (r0) participates in the mesencephalic caudal vesicular evagination (Figure 2a,b,d–f). Expression of *Fgf15* is weak at the rostral half of the midbrain, but it increases gradientally caudalwards to a maximum just in front of the IMB, at the expected locus of the m2 mesomere (Figure 2g). In contrast, the *Fgf8* signal is already restricted at this stage to the rostral evaginated part of the isthmus, behind the IMB, at the caudal pole of the midbrain vesicle (Figure 2h,i). The *engrailed* genes are held to be activated downstream of *Fgf8*. A double gradiental expression of *En1* overlaps with the isthmic organizer and characterizes the neighboring areas rostral and caudal to the IMB, penetrating maximally about one-third into the caudal basal midbrain (Figure 2j) and less so dorsally (Figure 2j–l). There is a specular *En1* gradient extending into the rostral hindbrain, possibly reaching the rh1 rhombomere (Figure 2j–l). The latter is labeled differentially by the *Otp* probe, contrasting with the *Otp*-negative isthmus domain (rh0, rh1; Figure 2m).

*Ascl1* shows rather widespread expression in the E11.5 midbrain ventricular zone, with a decreasing dorsocaudal gradient and some positive mantle ventrally (Figure 2n,o). Remarkably, its signal is sharply absent at a slightly evaginated part of the caudal midbrain neuroepithelium that covers the classic transverse ‘sulcus intraencephalicus posterior’ (Figure 2n,o). This sulcal area lies just rostral to the *Fgf8*-positive isthmic domain (compare Figure 2h,i), and exhibits a weak *Ascl1* signal in its ventricular zone (Figure 2o; note this is a topological horizontal section). If we check the *Otx2* signal again in similar horizontal sections (Figure 2d–f), it seems to occupy integrally the sulcal radial domain next to the IMB, thus making it a candidate for the postulated m2 unit at E11.5 (note Palmgren [17] used this landmark to identify the atrophic outpouching of m2, all the rest being m1). Nevertheless, we will see below that at later stages *Ascl1* comes to positively label the m2 domain.

A transverse negative gap apparently coinciding with the m2 locus and the associated sulcus at the caudal end of the midbrain vesicle is shown as well by *Gfra1* and *Id2* (Figure 2p,q). Other expression patterns, such as those of *Id4*, *Irx1* and *Irx2*, occupy the rostral midbrain and the rostral hindbrain (including the isthmus), but leave an interposed caudal midbrain gap devoid of signal, which may correspond to the prospective m2 (Figure 2r–u). Several *Lhx* genes (e.g., *Lhx1*, *Lhx2*, *Lhx5* in Figure 2v–z,aa) as well as the genes *Nkx2.2*, *Sfrp1,* and *Rprm* (Figure 2bb,cc,dd) label either large or limited parts of prospective m1 (in the mantle or ventricular zone) but leave out the caudal sulcal midbrain domain (*Lhx1* and *Lhx5* both distinctly label the isthmus; Figure 2v,y,z,aa). Interestingly, *Lhx5* labels the midbrain mantle at an intermediate rostrocaudal level, leaving negative both the rostral pole and the caudal sulcal area (Figure 2y,z,aa). It is unclear whether this early labeling is restricted to the basal plate (tegmentum), as is suggested in Figure 2y. In any case, at later stages this marker also becomes a selective alar m2 marker. The *Nkx2.2* gene, which otherwise is expressed along the whole forebrain (hypothalamus, diencephalon, and midbrain) as a thin longitudinal band approximating the alar-basal boundary [28], ends its longitudinal route at the caudal midbrain, where the expression domain is deflected dorsalward into the illustrated area (Figure 2bb) just in front of the caudal sulcal domain. Derived cells will later appear in the inferior colliculus.

The caudal sulcal ventricular zone of the midbrain appears densely labeled by the *Sall3* signal at several lateromedial parasagittal section levels (Figure 3a–d); this marker is also expressed in more rostral basal mantle elements in m1, but not in the adjoining isthmus domain. The latter is strongly labeled by *Sfrp1*, clearly leaving negative the adjacent caudal sulcal midbrain area (Figure 3e,f).

*Pax2* and *Pax8* both seem weakly expressed in the isthmic domain, stopping rostrally at the IMB, just caudal to the negative sulcal area (Figure 3g,h). In contrast to findings in the chick [15], m2 seems devoid of *Pax2* in the mouse, at least at this stage. In contrast, the *Pax5* signal extends rostralward from the positive isthmus and penetrates the caudal part of the midbrain (Figure 2i–k) analogously, as observed above for En1 (Figure 2j; note the positive sulcal area). *Pax3* shows alar expression in the midbrain and cerebellar hindbrain but leaves an area devoid of expression coinciding with the caudal sulcal locus (Figure 3l).

*Tal1* and *Tal2* show at E11.5 strong expression in the midbrain basal domain (ventricular zone and mantle; *Tal2* also has rather weak expression at the alar domain), but neither is expressed at the caudal sulcal area (Figure 3m–p). *Sst*-positive postmitotic cells are observed rostrally in the basal midbrain as well as in the basal isthmus, but are lacking at the caudal sulcal area (Figure 3q). The *Trh* signal appears selectively in a semilunar postmitotic mantle area that might correspond to the caudal sulcal area (Figure 3r).

Ventricular zone expression of *Wnt1* appears at E11.5 along a transverse line aligned precisely with the IMB, that is, in between the evaginated *Fgf8*-positive isthmus and the sulcal midbrain area (both being negative to *Wnt1*; Figure 3s,t). The *Wnt1* signal also extends rostralward along the midbrain roof plate, but not along the isthmic one. No relevant *Wnt8* signal was observed at the preisthmic domain (Figure 3u; there is some basal signal in m1, though). There is little *Tcf7l2* signal at either the isthmus or midbrain at E11.5 (Figure 3v), but at E12.5 positive mantle cells have started to aggregate subpially, particularly in an intermediate domain of the midbrain lying rostral to the sulcal area (Figure 3w); this primordium later seems to transform into the inferior colliculus (see below). *Sema3f* shows expression concentrated at the midbrain basal plate, but evades the caudal sulcal area (Figure 3x).

### 3.2. Midbrain and Isthmic Gene Patterns in E13.5 Mouse Embryos

At E13.5 the major structural primordia of the alar midbrain mantle are clearly distinguishable. Sagittal sections that reacted in situ for *Otx2* show a positive and elongated dorsorostral area that corresponds to the superior colliculus (SC; this apparently includes the still more rostral ‘tectal gray’ formation–TG—which we will point out only occasionally) (Figure 4a,b). The SC area limits ventrocaudally with a mantle domain devoid of an *Otx2* signal, which represents the inferior colliculus—IC—and its rostral continuation in the interstitial bed nucleus of the brachium of the inferior colliculus—the BIC (Figure 3a,b). Ventrocaudal to this band there appears again *Otx2* expression in what we interpret as the preisthmus or m2 derivative (Figure 3a,b). This domain seems wider dorsally (i.e., label ‘tor’, close to the roof plate), where it seems to participate somehow in the toral area of the inferior colliculus (more on this below). As the domain thins out ventralwards (topologically) it displays a sharp boundary with the *Otx2*-negative isthmus domain (across the IMB).

Examination of a dorsoventral series of horizontal *Otx2* sections (Figure 4c–g) identifies again these three alar midbrain territories (positive SC, negative IC, and positive preisthmus) and the molecular limit with the isthmic hindbrain, which partly evaginates into the midbrain vesicle (as seen in Figure 4d–g). We now notice that the IC is *Otx2*-negative only in its mantle zone, whereas its underlying ventricular zone is actually positive, thus establishing the validity of the marker for all the midbrain (the same pattern was discovered in the avian auditory midbrain [15]). These sections also show the sectioned caudal sulcal area. The posterior intraencephalic sulcus now forms dorsally a deep ventricular diverticulum that is not limited here by the isthmus domain (which lies more ventrally) and lies just *caudal* to the IC; it gradually flattens out ventralwards as the IC acquires a larger mass (Figure 4c–g). The sulcus is separated dorsally (sections in Figure 4c,d) from the median roof plate by an undifferentiated stretch of positive neuroepithelium that diminishes in surface extent ventralwards, finishing as a narrow radial interstice (the isthmus starts to appear at level of Figure 4d). This sequence corresponds to the change in dimensions of the alar preisthmus, which we already observed in the sagittal sections. We will see at later stages that the preisthmus seems to contribute to inferior collicular structure via this as yet undifferentiated dorsomedial ‘toral’ domain (the name alludes to its subsequent intraventricular bulge).

At the magnification represented it is hardly possible to distinguish that a stream of *Otx2*-positive cells exits the caudalmost midbrain. These migrate tangentially caudalwards along a cell-poor intermediate stratum of the isthmus (tiny dark dots in Figure 4e,f; see the Allen Developing Mouse Brain Atlas). We believe this represents the caudal migration of cells of the mesencephalic nucleus of the trigeminal nerve (mes5), which targets the isthmus and rh1 alar domains [20]. Various other studied gene markers label these particular neurons. On the other hand, we cannot identify at this stage any of the diverse alar derivatives of the isthmic region, though their development seems quite advanced.

In the second and third rows of Figure 4, we show the *En1* pattern at E13.5 (Figure 4h,i,l–s). We intersected two relatively equivalent horizontal sections labeled for *Lhx5* and *Ascl1* (Figure 4j,k respectively) because they are directly comparable to the overlying *Otx2* sections in Figure 4d,e. Sagittal parasagittal sections at lateral, intermediate, and medial section levels minimally illustrate the complex *En1* pattern (Figure 4h,i,l). The labeled preisthmus shows a strong subpial signal, a moderate to weak signal at intermediate levels, and a strong signal medially, the latter in apparent relation to the generation and migration of dopaminergic neurons of the ventral tegmental area and substantia nigra (compare Figure 4r,s). The dorsoventral series of horizontal sections shown in Figure 4m–s illustrate other details. The *En1* signal decreases gradientally across the preisthmus, barely reaching the caudal sulcal area (Figure 4m,n).

Turning now to Figure 4j,k, it will be noted that at E13.5 *Lhx5* distinguishes (similarly to *Otx2*) positive SC and preisthmic mantle domains (including the caudal sulcal area), whereas the IC mantle and ventricular zones are largely negative; the isthmus is also *Lhx5*-negative (Figure 4j). A similar pattern is shown by *Ascl1*, whose signal is restricted to the SC and preisthmic ventricular zones, including the sulcal area (Figure 4k). A much weaker signal slightly labels the IC and isthmic ventricular zone.

The fourth row in Figure 4 shows similar lateral parasagittal sections illustrating, first, restricted preisthmic expression of *Lhx1* (Figure 4t). Secondly, two sections showing massive complementary labeling by *Meis2* of the isthmus and neighboring hindbrain domains, as well as weaker labeling of the SC and IC/BIC midbrain domains; a distinct basal patch of mantle cells in m1 and the negative preisthmus stand out (Figure 4u,v). Finally, we see selective preisthmic labeling (with separate signal in basal m1) with the *Pitx1* probe. Note the constancy and sharpness of the IMG boundary, irrespective of the side labeled.

*Gad1* and *Gad2*, both markers of GABAergic neurons, are strongly represented at the preisthmic level (not so at this stage at the IC primordium; Figure 4x,bb). The *Gad1* signal also appears in the SC primordium. Dorsal and ventral intersections of the semicircular arch of *Fgf8* expression along the rostralmost isthmus (=the isthmic organizer) are seen in Figure 4y. Note the implicit topographic obliquity of the *transverse* IMG boundary. The *Fgf15* signal appears as a positive transverse line that in comparison to the *Gad1* pattern seems to separate the negative preisthmic domain from the IC domain (Figure 4z,aa).

The preisthmic midbrain territory is clearly invaded by the expression of *Pax5* (Figure 4cc,dd,ee). This pattern is also detailed in a series of horizontal sections (Figure 4a–g). Note that both the isthmus and preisthmus are variously labeled, but the IC area remains negative. A similar preisthmic *Pax8* labeling was observed (Figure 5h). Comparable selective positive patterns are likewise provided by *Penk*, *Sst*, *Pou6f2*, *Sox14,* and *Six3* (Figure 5i–n, respectively). Note in contrast the selective preisthmic *negativity* demonstrated by *Pou3f3* (Figure 5o).

A series of parasagittal sections labeled with *Tal1* illustrates its marked expression laterally at the alar preisthmus (Figure 5p), as well as more ventrally along the transition into the preisthmic basal domain, where it is separated distinctly from an analogous isthmic basal signal by a negative narrow tissue gap associated with the posterior intraencephalic sulcus (Figure 5q–t). A parasagittal section (Figure 4u) and a dorsoventral set of five horizontal sections (Figure 5v–z) labeled with *Tal2* illustrate major labeling at the preisthmus and SC, and a minor periventricular signal at the IC primordium. There is only very weak ventricular expression at the isthmus.

The last row of Figure 5 shows the *Tcf7l2* expression pattern in a parasagittal section (Figure 5aa) and five dorsoventral horizontal sections. This marker selectively characterizes the SC and IC mantle primordia, leaving negative the preisthmus and isthmus. Note the lack of signal at the caudal sulcal area.

The *Lhx2* pattern is shown in a set of six dorsoventral horizontal sections. It consists of labeled deep mantle cells at the SC, IC, and preisthmus (not so with the isthmus), with a substantial superficial mass of IC that remains devoid of expression, particularly in the more ventral sections (Figure 6a–f). Figure 5g illustrates a very lateral parasagittal section reacted for *Npy*. The signal is largely restricted to the preisthmus, though a number of positive cells seem to disperse tangentially across the BIC domain into the SC. The second row in Figure 6 displays a series of dorsoventral horizontal sections illustrating *Lhx5* expression (Figure 6h–n). We distinguish the positive SC and preisthmic domains (including the caudal sulcal area) and the intercalated negative IC domain. Note a subpial positive cell stream seems to invade the superficial IC mantle out of the preisthmus (Figure 6i–k). There is also expression of this marker at the isthmus. The boundary across the IMB with the preisthmus is visible at the levels of Figure 6h–l, partly thanks to the flattened posterior intraencephalic sulcus.

*Nkx2.2*-positive cell patches characterize the dorsal IC mass (Figure 6o–q). More ventrally there is more compact signal associated with the caudal sulcal area along the dorsoventral dimension of the preisthmus (Figure 6r,s). Finally, the last sections illustrate the confluence of the preisthmic transverse pattern with the longitudinal band coursing along the alar-basal boundary (Figure 6t,u). These last sections are cut essentially transversal to the midbrain and typically show radial ventral distribution of *Nkx2.2*-positive cells in the laterorubral tegmental area and dorsalward migration of a clump of periaqueductal positive neurons.

The *Wnt1* expression remains stable along the transverse IMB arch (two intersections are seen) and the midbrain roof plate (Figure 6v,w). *Pou4f1*, another marker of the migrated mes5 neurons, as well as a selective marker of the SC and IC mantle in m1 (not the preisthmus in m2), is shown in the parasagittal and horizontal sections (Figure 6x,y). Some positive mes5 neurons remain fixed along the migration pathway targeting rh1 (the rh1/rh2 boundary is not violated).

### 3.3. Midbrain and Isthmic Gene Patterns in E15.5 Mouse Embryos

At E15.5, *Ebf3* is a marker of the preisthmus and SC/TG, leaving the BIC/IC domain essentially negative except for a few dispersed cells. There is also a negative boundary with the isthmus where a patch of signal appears in the mantle (Figure 7a). In more medial parasagittal sections, extensive labeling of the midbrain periaqueductal gray formation is observed (Figure 7b). The *Ascl1* signal appears only at the ventricular zone of the preisthmus and SC, leaving the IC and isthmus domains negative (Figure 7c,d).

The preisthmus (with or without the SC) appears selectively labeled by *Nhlh2*, *Lhx5*, *Foxp4*, and *Gata2* (Figure 7e–h). *Gata2* shows a linear pattern of expression at the boundary of the preisthmus with the IC that recalls the pattern of *Fgf15* at E13.5. These sites can be easily ascribed to the preisthmus (and SC) by comparison with *Otx2* expression in three lateromedial parasagittal sections (Figure 7i–k). We confirmed that the IC keeps *Otx2* expression in its ventricular zone (Figure 7k,l). A paramedian parasagittal section also illustrates the *Otx2* signal in the basal preisthmus, representing a quite narrow transverse and radial interstice placed in a retrorubral position (Figure 7l). Four dorsoventral horizontal sections (note there is left–right obliquity of these sections; both sides are of interest) labeled likewise for *Otx2* illustrate the wide dorsal extent of the positive alar preisthmic domain (next to the negative IC) at the level where it makes contact with the local roof plate (Figure 7m). More ventrally, the alar preisthmus gets reduced to a narrower *Otx2*-positive band that underlines the caudal aspect of the IC and limits caudally with the negative isthmus domain (Figure 7n–p). It is appreciated in these images (particularly on the left side) that at E15.5 the isthmus has penetrated significantly topographically into the midbrain vesicle (intussusception by morphogenetic deformation of the IMB). The four underlying patterns of the markers *Gad2, Gata3, Id4,* and *Tal1,* identify specifically the preisthmus and show a curved boundary with the isthmus, with development of a marked convexity towards the midbrain (Figure 7q–t).

A set of four parasagittal sections next illustrate the *Meis2* pattern, which is negative in the preisthmus and most of the SC but positive in the BIC/IC complex (Figure 7u–x). Note also that the dorsal rim of the latter domain shows a stronger signal than its ventral rim, as is observed particularly in Figure 7w. This corresponds to the incipient differentiation along this more strongly labeled locus of the separate *intercollicular domain* (ICol), intercalated between the SC and IC. There is also a separate patch of *Meis2* expression in the basal m1 area. The isthmus and other hindbrain prepontine areas are also strongly positive.

The superficial preisthmic mantle stratum begins to differentiate into what we call the ‘subbrachial nucleus’ (SubB; [21]). This primordium adopts a characteristic curved shape above the isthmic superficial elements (due to the curving of the IMB). It shows a *ventral* head-like thickening (oriented topographically rostral) and a longer *dorsal* tail portion (oriented topographically caudal). We show this characteristic image displayed by four different gene markers (*En1*, *Pax5*, *Pax8*, *Pou6f2*) in Figure 7y,z,aa,bb.

The preisthmus is delineated perfectly in negative by the markers *Irx1*, *Sst*, *Irx2*, and *Lhx9*. All of them label distinctly parts of the isthmic mantle at rh0 (well delimited from rh1), as well as the BIC/IC complex (Figure 8a–d). In horizontal sections, *Foxp2* shows deep to superficial gradiental labeling of the IC primordium plus a weaker SC signal. There is also some separate weak signal at the intermediate stratum of the preisthmic mantle, caudal to the IC (Figure 8e–h). The positive patch at the left bottom corner of Figure 8h is an unidentified superficial isthmic mass. Note that at E15.5, the midbrain caudal sulcal area is hard to identify due to its progressive flattening.

Another good marker for the IC is *Tcf7l2* (Figure 8i–l). The collicular labeling occupies the whole mantle, while more rostrally, possibly at levels through the BIC nucleus, this mass appears separated by a negative periventricular stratum from the corresponding positive ventricular zone. The SC shows a wholly different pattern and the preisthmus is essentially negative except for a few periventricular cells. Similar sections labeled for *Lhx2* illustrate in contrast a negative IC core nucleus with an underlying positive periventricular stratum, which is continuous with the preisthmic mantle (Figure 8m–p; here the preisthmus reaches the roof plate in Figure 8m,n) as well as with the SC counterpart. The superficial mantle intercalated between the negative IC and SC domains shows an increased signal of *Lhx2* (ICol; Figure 8m–p); this transitional area corresponds to the incipient intercollicular area introduced above with *Meis2*. The IC domain appears fully and selectively negative with *Lhx5*, whereas the SC is weakly labeled in a diffuse pattern (Figure 8q–t). In contrast, the preisthmus selectively shows its characteristic mantle signal (the isthmus being wholly negative). The preisthmic domain dwindles in extent ventralwards.

The gene *Six3* shows positive cells distributed through lateral parts of the BIC/IC complex, but is otherwise more strongly represented along the preisthmus and is absent from the isthmus and SC regions (Figure 8u–x).

The last row in Figure 8 is dedicated to examples of markers selectively labeling the basal or tegmental preisthmus, namely *Zfhx3*, *Sst*, *Pou6f2*, and *Nts* (Figure 8y,z,aa,bb). The differential distributions of the labeled elements suggest that various neuronal types are more or less mixed within the narrow preisthmic tegmentum. We do not show the local dopaminergic or serotonergic neurons reported by our group [21,27].

### 3.4. Midbrain and Isthmic Gene Patterns in E18.5 Mouse Embryos

The first row in Figure 9 presents the E18.5 *Otx2* pattern. As before, the SC and preisthmus domains stand out, while the largely negative BIC/IC complex only appears labeled at its ventricular zone (Figure 9a–d).

The lateral parasagittal section in Figure 9a illustrates the characteristic headed worm shape of the subbrachial nucleus (the superficial stratum of the preisthmus), as described at E15.5. The three horizontal sections in Figure 9b–d show a sequence through the positive preisthmus, passing from its enlarged dorsal part to its much narrower ventral portion. Note the posterior intraencephalic sulcus is recognizable at levels of Figure 9c,d and only part of its ependymal ventricular zone expresses the *Otx2* marker. This indicates that the bottom of the sulcus may represent the IMB, so that its rostral lip belongs to m2, while the caudal lip is isthmic (representing the organizer). A number of positive preisthmic cells seem to violate this boundary and disperse in the neighboring isthmic mantle (Figure 9c,d). The IC core is negative and appears surrounded deeply and superficially by a dispersed shell of positive cells. The negative deep stratum lying under the deep IC shell corresponds to the local periaqueductal gray (PAG). Labeling in the SC is restricted to intermediate and superficial cell layers, plus the ventricular zone. Its own part of the PAG is also negative.

Interestingly, the dorsalmost *Otx2* section shown (Figure 9b) illustrates dorsomedial continuity of the positive preisthmic mantle with symmetric bulges at the dorsocaudal midbrain midline. We interpret these bulges as derived from the dorsomedial toral undifferentiated domain distinguished at the same position at the preceding stages. These bulges emerge between E15.5 and E18.5, indicating late growth. Their significance will be considered in the Discussion. We will explore their genoarchitectonic profile and histogenetic evolution here and in the next section.

The second row in Figure 9 shows the *Calb2* (calretinin) and *Cbln2* patterns in parasagittal sections (Figure 9e–h, respectively). *Calb2* appears selectively in the subbrachial preisthmic nucleus (a subpial formation), showing its typical head-tail configuration (Figure 9e). A deeper section interests the intermediate preisthmic stratum, forming the incipient cuneiform nucleus (Figure 9f). The isthmus and BIC/IC domains are negative. In contrast, *Cbln2* leaves unlabeled the preisthmus and selectively marked the BIC/IC/ICol complex. Note the ICol area appears more intensely labeled than the underlying BIC/IC. The SC appears labeled in several layers and can be perfectly distinguished in Figure 9h from the massively labeled rostral tectal gray (TG) formation (another midbrain retinorecipient locus; see [26]).

The basal or tegmental preisthmus appears distinctly labeled with *Dlk1* (Figure 9i). The underlying Figure 9p shows a remnant of *Fgf8* activity at a ventricular locus supposed to be just caudally adjacent to the ventricular projection of the basal preisthmus in Figure 9i.

We examine again the *Ebf3* pattern at E18.5 (centered on the preisthmus and the SC, plus isolated isthmic cell masses; Figure 9j), as well as the *Gad1* pattern (Figure 9k), which illustrates the presence of GABAergic neurons in the preisthmus BIC nucleus and IC shell, and some layers of the SC. The isthmus proper remains unlabeled. *Gata 3* (Figure 9l) weakly labels the preisthmus and faintly labels the ICol area.

The pattern of *Gfra1* is shown in six lateromedial parasagittal section levels (Figure 9m–o,q–s). The major labeled domain is the preisthmus, always found under the negative BIC/IC complex. The three more lateral sections interest the subbrachial and cuneiform nuclei, whereas the medial sections illustrate the preisthmic PAG area and end at the dorsomedial toral region. The superficial and intermediate strata of the SC are positive, and it can be observed that the rostral tectal gray visual formation is wholly negative with *Gfra1* (TG; Figure 9m–o,q–s).

The last row in Figure 9 displays four dorsoventral horizontal sections labeled with *Lhx2* (Figure 9t–w). These sections illustrate the dorsomedial continuity of the positive preisthmic mantle with the symmetric toral bulges that have emerged out of the paramedian roof area, just in front of the isthmus. *Lhx2* distinctly labels the ventricular zone and part of the mantle stratum of the toral bulges. Their protrusion into the midbrain ventricle causes the tip of the toral bulges to appear tangentially sectioned in Figure 9v. The last section in Figure 8w shows a standard view of a more ventral part of the IC and the caudally adjacent labeled preisthmic territory.

The described toral phenomenon was further studied by examining the *Foxp2* marker, which does not label the preisthmus (and the ICol area) and strongly labels the IC core and the BIC nucleus, apart from multiple layers in the SC. A series of eight dorsoventral horizontal sections (Figure 10a–h) show that the IC core extends in a hook pattern, passing above the preisthmus and into the dorsomedial toral area next to the roof plate. Moreover, *Foxp2* also labels the mantle of the toral bulges, in continuity with the IC core (Figure 10a–d). There is, though, a sand-clock-like shape with an interposed, less-labeled constriction corresponding to the IC-preisthmic boundary (Figure 10b–d). The toral mantle disappears progressively into Figure 10e,f, where mainly the ventricular lining is cut tangentially. The last section shows the positive BIC nucleus, which continues rostralward toward the IC core, limited dorsally by the negative ICol area. The last three sections show some positive preisthmic patches caudal to the IC occupying a superficial stratum (subbrachial nucleus), as well as the caudally adjoining positive parabigeminal isthmic nucleus (SubB; PBG; Figure 10f–h).

Indeed, the isthmic subpial PGB nucleus can be now recognized as a group of superficial patches. These patches are labeled distinctly with *Enc1*(Figure 9i) or with *Meis2* (Figure 9j), apart from other, deeper neurons (Figure 10k–n). This pattern clearly distinguishes the isthmic mantle from both the preisthmus and rh1 (Figure 10i–l). Note the better developed, higher *Meis2* signal of the ICol contrasting with the BIC/IC complex (Figure 10k–o). The toral bulge seen in Figure 10p also strongly expresses *Meis2*.

In the last row of Figure 10, we show four similarly lateral parasagittal sections illustrating the superficial preisthmic signals of *Pax5*, *Penk*, *Npy*, and *Zeb2* (Figure 10q–t). The *Npy* signal obviously is also present in the IC, though this may represent migrated cells.

Dorsoventral sections displaying *Nrp1* expression reveal labeling of intermediate and superficial strata of the preisthmus, with less signal at the corresponding PAG portion (Figure 11a–d). On the other hand, the PAG under the SC seems specifically labeled, similarly to the IC shell.

We next present images of the *En1* signal to clarify sequential changes at the dorsomedial toral area. At E16.5 (Figure 11e), the toral area is positive jointly with the preisthmus (not so the IC), but remains undifferentiated, hardly starting to bulge into the ventricular space. At 17.5 (Figure 11f), the toral bulges are more evident and show the *En1* signal at their mantle layer. At E18.5 (Figure 11g,h), the dorsomedial toral bulges are fully formed, protruding strongly in the ventricle and maintaining *En1* expression. Further evidence on these formations was found in the *Tal2* expression pattern at E18.5, illustrated in one sagittal section (Figure 11i) and three horizontal sections (Figure 11j–l). This marker mainly labels the ventricular zone of the toral bulges, being absent from both the IC and the interposed preisthmic territory.

We also examined the *Tcf7l2* signal at E16.5 (Figure 11m–p), E17.5 (Figure 11q–s), and E18.5 (Figure 11v–y). At E16.5 and at section levels above the preisthmus, this marker is already continuously expressed from the IC core into the immature dorsomedial toral area, leaving the roof plate negative (Figure 11m,n). Once the series intersects the preisthmus more ventrally, this appears negative, though a remnant of the positive toral area is still present (Figure 11o,p). A similar pattern was observed at E17.5 (Figure 11q–s) as well as at E18.5 (Figure 11v–y). This pattern suggests that the toral area is a dorsocaudal prolongation of the IC domain, which ends near the dorsal part of the preisthmus. However, the selective labeling of the toral area by markers such as *En1* and *Tal2* suggest that there are aspects of its molecular profile comparable to the preisthmus that are absent at the IC, irrespective of shared *Tcf7l2* expression.

The free space at the end of row five in Figure 11 was used to insert two views of selective labeling at the basal tegmental preisthmus with *Slc6a3* and *Sst* (Figure 11t,u). A singular basal preisthmic dot-like area of expression seen periventricularly in sagittal sections labeled with *Trh* led to an examination of the coronal sections and the finding that there is a *Trh*-positive distinct cell group in the ventral PAG associated with the preisthmus, out of which less numerous similar cells spread out laterally and ventrally (Figure 11z,aa). Finally, both *Zfhx3* and *Zfhx4* were found to selectively label isthmic superficial masses and other rh1 cell masses, as well as the BIC/IC complex, leaving the preisthmus domain devoid of significant expression (similar to *Enc1* and *Meis2* described above).

### 3.5. Midbrain and Isthmic Gene Patterns in Postnatal Mice

Figure 12, corresponding to postnatal results, shows uniformly adult brain material. The first row of images is dedicated to the basal preisthmus, represented by an elongated and narrow tissue band at the caudal end of the basal midbrain where cells expressing selective markers such as *Calb1*, *Cck*, and *Sst* are found, among other markers such as dopamine and serotonin (Figure 12a–c). The two first images clearly show the dorsal and ventral tegmental decussations, which characterize the m1 ventral midline or floor, placed just in front of the m2 floor. Behind m2 we see the IMB limit and at some distance the decussation of the brachium conjunctivum, known to decussate across the isthmic floor ([8]; this is irrespective of the fact that some outdated sources place it in the midbrain). Figure 12a,b also distinctly show the prepontine interpeduncular nucleus complex lying just caudal to the superficial end of the basal m2 at the apex of the cephalic flexure (see [29]), as well as the major mass of the dorsal raphe nucleus periventricularly (dr; see [27]).

The second and third rows show coronal sections representing actually horizontal sections across the IMB and the preisthmus, labeled respectively with *Cbln2*, *Dach1*, *Gfra1, Kctd4,* and *Penk* (Figure 12d–i). *Cbln2* is a marker of the superficial isthmic and rh1 nuclei, which labels as well parts of the IC, but leaves the intercalated preisthmic domain wholly negative. This lack of signal can be followed from the PAG across the intermediate cuneiform area into the superficial derivative, the subbrachial nucleus (PAG, Cnf, SubB; Figure 12d). *Dach1* weakly labels a few neurons distributed across the three strata of the preisthmic domain and leaves unlabeled the IC and the isthmus (Figure 12e). *Gfra1* is a preisthmus marker and the corresponding sector of the PAG appears densely labeled (note the boundary with the IC PAG sector), as well as the adjacent Cnf nucleus (Figure 12f). There are also some weakly positive isthmic cells. *Kctd4* selectively labels preisthmic elements and illustrates here mainly the preisthmic PAG and the adjacent Cnf nucleus (Figure 12g). The last two sections in Figure 12h,i show the *Penk* signal at the preisthmic domain (again PAG and Cnf) as well as at the underlying superficial isthmus (the PBG nucleus).

Returning to the dorsomedial toral area, we first show it at P1 labeled selectively in connection with the preisthmus with *En1* (Figure 13a–c). Labeling is massive at the toral region and decreases somewhat into the adjacent preisthmus. There is no signal in the IC core. Complex additional details characterize the isthmic derivatives that express *En1*, a marker that is also present at the diverse sectors of the PAG (Figure 13a–c). Next, we examined the *Tcf7l2* signal at the IC and the toral region at P1 (Figure 13d), P2 (Figure 13e), and P4 (Figure 13f). We observed over these stages that the toral bulges gradually become flattened. Sagittal sections at P4 labeled with *Tcf7l2* or *Six3* show the marked flattening of the former toral bulges (tor), practically changing into the thin roof-related membrane that separates the rostral cerebellar lobules from the midbrain ventricular cavity (Figure 13j,m).

The sagittal sections shown in Figure 13g–i correspond to the selective labeling of preisthmic patches of PAG with the markers *Slc17a8*, *Id4*, and *Rgs4*, respectively. The sagittal images in Figure 13k,l illustrate superficial labeling of the subbrachial preisthmic nucleus (SubB) with *Penk* and *Pitx1*, respectively. Other markers also selectively labeling the SubB nucleus are *Ebf3* and *Nts* (Figure 13n,o), whereas *Irx1* labels the complementary isthmus and BIC/IC domains (Figure 13p).

## 4. Discussion

Our purpose was to clarify the delimitation, development, genoarchitectural differentiation, and final anatomy of the postulated preisthmic domain, which was originally related to the m2 prosomere of the caudal midbrain by Hidalgo-Sánchez et al. [15]. The specific historic antecedents of this notion were presented in the Introduction, but a few general comments are nevertheless convenient.

### 4.1. The Theoretical and Technical Background of Present Studies

The anatomic transition between the midbrain and the pons, thought to be in direct contact in the nineteenth and twentieth centuries, progressively became a battleground between (1) the traditional premolecular ‘columnar’ studies, which were not interested in having a series of transverse neuromeric subdivisions (the ‘anatomically important’ aspects were held to be functional and contained under the concept of longitudinal sensory and motor neuronal ‘columns’ [30]), and (2) the soon heterodox neuromeric ideas represented initially by, e.g., [31,32], and later by [17,18,33]. These authors did not know about neuromeric functions (we are still learning about them), but they had *seen* that the early embryonic hindbrain of all vertebrates appears divided into a series of transverse neuromeric bulges called *rhombomeres*, and the midbrain shows one or two *mesomeres*. This scenario saw the midbrain-pons transition in a more elaborated manner, resolved nowadays in the concepts of a preisthmic m2 segment caudal to m1, followed by a hindbrain *prepontine* area divided into the isthmus (rh0) and rh1 neuromeres [22,23,34]. The important boundary, in more ways than one, was not the imaginary midbrain-pons limit, but the isthmo-mesencephalic or m2-rh0 limit, to which secondary organizer properties and crucial evolutive intertagmatic roles were later attached (review in [12,35]).

The neuromeres are not functional mechanisms, but *generative units* with repetitive modular dorsoventral organization (metamery), the sites of distinct patterning, proliferative, migratory, and differentiative phenomena. This morphologic insight could have been regarded as complementary to the columnar one, also in functional perspective (see discussion in [20,34]), but for some reason they were considered mutually incompatible, and the non-functional neuromeric phenomena were simply left aside for 100 years. Kuhlenbeck [36,37] concluded that neuroepithelial neuromeres were early transverse embryonic subdivisions of the neural tube wall that disappeared without consequences as the really important postmitotic neuronal columns emerged in the brain wall. No one asked what mechanism patterns the columns along the DV and AP dimensions; they notably produce different nuclei at different rostrocaudal levels of the hindbrain, and considerable uncommented diversity exists in the postulated forebrain columns. This was the orthodox scenario in the field for many decades until developmental genes made their appearance and that issue was first addressed.

Once molecular biology progressed sufficiently, it was possible to map the expression domains of single genes in the developing brain. It was soon discovered that developmental genes admitted both longitudinal and transversal expression patterns, which indicated that the neural wall was exposed to overall dorsoventral and anteroposterior patterning. Molecular regionalization mechanisms were studied in much detail thereafter, partly through the notion of secondary organizers. This opened the way to a synthesis of columnar and neuromeric causal mechanisms, and the forgotten neuromeres duly returned to the literature around the mid-eighties. Neuromeres do not need to disappear so that columns emerge. There are genes for neuromeric patterns and genes for columnar patterns (often downstream ones). The columnar neurons must be born out of neuroepithelial progenitors positioned within specific neuromeric molecular coordinates. Segmented progenitors surely have something to say about what becomes what in the brain. As a consequence, neuromeric structural patterns persist in the adult, delimited by cryptic molecular boundaries. Moreover, mantle columns *need* to subdivide into different parts if we want to register and understand the real complexity of resultant adult structure. The whole hypothalamus and the whole dorsal thalamus were originally held to be *uniform* columns, but obviously cannot be understood functionally or explained causally without partitioning them. The issue is that neuromeric patterns and boundaries are needed for that level of analysis (e.g., [37,38]).

In any case, an early result of the molecular era was that some genes permanently mark the interneuromeric boundaries or are restricted to single neuromeres or neuromere parts (which accordingly do not disappear physically over development; they just become macroscopically cryptic as differentiation and histogenesis advance). Nowadays we can label selectively the entire progeny of given neuromeres in the adult brain by means of selective transgenic design (e.g., [8,39]). We also can see how many neuromeric units participate with specific dorsoventral modules in a functional column [40]. We have a clear idea of the neuromeres and columns we can find in the hypothalamus [41].

After some 40 years of progress in these aspects, it seems odd that there still exists a neuromeric unit—the m2 mesomere that occupies us here—that is largely unknown by embryologists and anatomists, even though it was first postulated 100 years ago [17] and glimpsed even before [32]. After studies of the midbrain in several vertebrates, Palmgren [17] concluded that practically the whole midbrain originated from the rostral large m1 mesomere. Contrarily, he thought that the minor caudal embryonic unit (m2) remained atrophic, showed little growth in surface, and produced no neuronal derivatives. It was thus more a boundary of m1 with the isthmic hindbrain than a proper generative unit for a particular part of the brain. This strange concept was later assumed by Vaage [18] in his general review of chicken neuromeres, as well as in his first-rate descriptive work on the developing isthmic region [13]. We also reluctantly used this unsatisfactory notion in [14].

Eventually, Hidalgo-Sánchez et al. [15] applied genoarchitectonic procedures to the analysis of the origin of chicken isthmic nuclei. Vaage [13] had credibly concluded that one of them was mesencephalic, others were isthmic, and still another could be perhaps ascribed to rh1; these conclusions were also supported by our autoradiographic neurogenetic data [14]. Having read about the role of *Otx2* as defining the caudal boundary of the midbrain, we decided to map the position of the problematic ‘falsely isthmic’ midbrain nucleus relative to the *Otx2* boundary, adding other local markers such as *Fgf8*, *Gbx2*, and *Pax2*. These data revealed that indeed the nucleus was molecularly mesencephalic, that is, *Otx2*-positive. It lay rostral to the *Otx2*-negative isthmic derivatives, but, surprisingly, caudal to the structurally and molecularly distinct auditory structures (whose mantle uniquely downregulates *Otx2*; we presently found this feature reproduced in the mouse). We noticed that some other structures seemed to be in the same situation and could be jointly labeled selectively with *Pax2* (not so with the auditory midbrain and the rest of m1).

These data, connected with the classic m2 concept, allow the *novel conclusion* that our *Otx2*/*Pax2*-positive domain caudal to the torus had to be the missing brain domain derived from the m2 mesomere [15]. It had been long misidentified as a part of the isthmus (which non-neuromeric authors believed a part of the midbrain). The m2 thus became a normal neuromere, a distinct generative unit, even if still surprisingly small when compared with m1.

### 4.2. The Preisthmus or m2 Mesomere

We built here upon the earlier work in the adult mouse [21] (note the first author in this publication was E.Puelles, not LP). We basically expanded the developmental knowledge about the mouse preisthmus. As far as we are aware, nobody else has approached this subject in any detail, with the exception of Hidalgo-Sánchez et al. [12].

Our present results in the mouse corroborated the old observations of Palmgren and Vaage [17,18] on the early existence of a caudal small diverticule of the mesencephalic vesicle, already known to classic authors as the ‘posterior intraencephalic sulcus’ of von Kupffer [32]. Palmgren and Vaage thought it represented the small and supposedly atrophic cavity of the m2 mesomere. We showed that this diverticule is indeed included in the midbrain according to the *Otx2* criterium, though its caudal lip seems to belong to the *Fgf8*-positive isthmus. The transverse *Otx2*/*Fgf8* limit represents the modern molecular concept of the isthmo-mesencephalic boundary (IMB; [8]), which has been associated in much experimental work with the isthmic organizer. We have also confirmed previous results in the literature indicating that the isthmus does not stop rostrally at the early macroscopic constriction visible between the midbrain and the hindbrain, but that it penetrates slightly into the caudal aspect of the midbrain vesicle.

Another developmental gene thought to be relevant for the isthmic organizer—*Wnt1*—was found to be expressed precisely in a transverse ring configuration along the IMB limit (as well as longitudinally along the midbrain roof plate). The neurogenetic *Ascl1* marker showed an interesting exclusion of the sulcal m2 area from its ample area of midbrain expression. Several other genes (*Id2, Lhx2, Sfrp1, Pax3,* or *Irx1*) respected specifically the m2 territory at E11.5. In contrast, *Sall3* labeled nearly exclusively the m2. The *Lhx5* gene respected the sulcal m2 area at E11.5 but showed selective m2 labeling at all later stages. It is plausible that the early pattern of this marker refers to basal plate levels (the alar components being perhaps still absent), whereas the later m2 positivity includes an alar pattern. We could not verify that *Pax2* is expressed in the mouse m2 domain, in apparent contradiction to our positive data in the chick [15]

At E11.5 there are genes whose signal crosses the IMB and distributes gradientally in both the caudal midbrain and the rostral prepontine hindbrain (e.g., *En1*, *Pax5*). We noted at subsequent stages that the midbrain distribution of these markers remained largely restricted to m2 and did not significantly penetrate the auditory midbrain.

### 4.3. Anteroposterior versus Dorsoventral Dimensions and Regionalization in the Midbrain

The m1/m2 subdivision is an AP pattern, and this topology is permanent, irrespective of later deformations in the shape and partitions of the alar midbrain due to differential growth. Indeed, as development proceeds, three deforming phenomena occur in parallel.

On the one hand, the midbrain alar and roof plates grow much more in surface than the basal plate and floor areas. Within the alar plate, neurogenesis reportedly occurs in a main rostrocaudal gradient, with a secondary basodorsal gradient ([42]; but see [20]). On the other hand, the whole neural tube bends ventrally at the cephalic flexure, with an apex just under the midbrain, apparently due to the summed effect of all forebrain and hindbrain alar plate regions growing more that the basal ones. The third minor deformation reflects the partial sucking up of the isthmic hindbrain into the caudal aspect of the midbrain (a process that is already incipient at E11.5, as seen in the horizontal sections), and later causes a larger intussusception and mesencephalopetal arching of the IMB.

The combination of the first two processes of deformation generates the deformed wedge shape of the midbrain in its AP dimensions, with a very small floor area (practically restricted to the dimension of the root of the oculomotor nerve) which contrasts with extensive alar and roof plates (the degree of this deformation varies in different vertebrate species; [20]). Imaginary equidistant transverse lines traced from the floor to the roof would spread out dorsalward in a fan shape, representing in every case the transverse dimension. The rostrocaudal neurogenetic gradient explains that the caudal imaginary transverse lines spread out more caudalwards than rostralwards (proliferation persists longer at caudal levels).

The fact that as development proceeds the m1/m2 boundary (similarly as the IMB) increasingly bends backwards can thus be easily explained, because we deal here with one of the imaginary caudal transverse reference lines, which widely fans out caudalwards while retaining its topologic transverse character. Topographically, the caudal preisthmus thus apparently comes to lie ‘under’ the auditory BIC/IC complex, though *it really remains topologically caudal to it*. These topologic considerations are somewhat banal, but they need to be clear to evade pitfalls in developmental reasoning on gradients or on the interpretation of adult connectivity patterns, which usually are organized topologically rather than topographically.

Our analysis suggests that the auditory IC and BIC formations, irrespective of their common functional specialty, molecular similarity, and probable mutual interconnections, possibly belong to different AP domains of the m1 alar plate. The BIC formation can be easily understood as a ventrolateral subdomain of the superior colliculus AP domain. The IC is the single rostral neighbor of the preisthmus, so that both participate in parallel in the PAG.

The dorsocaudal tip of the SC is also much deformed caudalwards, obviously following the general fan-shaped alar deformation. If we add the rostral tectal gray (TG), also complemented ventrolaterally by the rostralmost BIC, we believe that alar AP regionalization in front of m2 may entail IC, SC/BIC, and TG/BIC transverse sectors, making with the preisthmus a total of four distinct AP midbrain domains (the role of the IColl domain needs further study). This analysis of midbrain AP patterns also affects the theory of PAG structure, presently still largely driven by DV subdivisions and columnar anatomic assumptions.

### 4.4. Development and Fate of the Preisthmic Neuromere

It may be wondered whether the preisthmus is a neuromere by itself, since it might be conceived simply as the caudalmost AP subdivision of m1 secondary to isthmic patterning. We know by now that neuromeres do not need to bulge out and be limited by constrictions to exist (the posterior intraencephalic sulcus thus is irrelevant). They can be cryptic from a macroscopic viewpoint [43], but real from a mechanistic molecular viewpoint [5,44]. The tectal gray, superior colliculus, and inferior colliculus of the midbrain are not considered neuromeres because these partitions (also molecular) only exist in the alar plate; they lack corresponding distinct basal parts. What establishes the neuromeric nature of an AP molecular delimitation in the brain is that the interneuromeric limits are general ones dorsoventrally, separating whole transverse units of the neural tube (the criterium of *completion* [44]). While we do not always know of genes that label a complete neuromere (both basal and alar parts; but see the case of rh4 [39]), in the case of m2 we have the *Sall3* pattern (Figure 3a–d). Moreover, we found some shared markers in both alar and basal parts of the preisthmus, as well as a dorsoventral continuity of its general boundaries from roof to floor. We thus postulate that m2 fulfills the criterium for representing a neuromeric unit of the midbrain, distinguished in various molecular and structural aspects from the larger m1 unit (e.g., there is no motoneuronal population in m^2^).

Hidalgo-Sánchez et al. [12] coherently inserted the preisthmus as the midbrain area closest to the isthmic organizer in their schema of territories affected by the signaling originated at the organizer (their Figure 10; see also our Fig.1). This implies that the molecular peculiarities of this territory probably result from their exposure to high levels of FGF8 and WNT1 in the context of an *Otx2-*positive territory and relatively high levels of other determinants (e.g., *Fgf15, En1, En2, Lhx5, Pax5, Pax8*; see full review in [12]).

It has been reported that neurogenesis is initially retarded selectively at m^2^, compared to m1 in general ([20]; data for rat, chick, and lizard). This effect may be related to *Her5* activity in this area (e.g., [45]; review [12]), as well as by the observed selective lack of proneural *Ascl1* expression at E11.5 (Figure 2n) and high levels of *Fgf15* and *Sall3* (Figure 2g and Figure 3a–d).

Both prenatally and postnatally, a significant set of positive and negative markers detect rather precisely the same boundaries of the preisthmus, either relative to the isthmus (marked by *Fgf8* and *Wnt1* signals) or to the auditory complex of m1 (IC; marked by its absence of the mantle *Otx2* signal and various specific positive markers, such as *Meis2, Pou3f3, Pou4f1*, and *Tcf7l2*). *Lhx5* was peculiar in strongly labeling a thick periventricular stratum and weakly a thinner subpial stratum that seems to extend tangentially into the marginal stratum of the inferior colliculus (Figure 6h–n). This suggests a migration of preisthmic cells into the superficial IC (the ulterior IC shell?). Other suggestions of tangential migration are raised by the patterns of *Npy*-, *Six3*-, and *Nkx2.2*-positive cells, which also seem to spread out of the preisthmus into the auditory complex.

At E15.5, we observed a more marked intussusception of the isthmic mantle into the mesencephalic lobe (Figure 7i–l). The two separate strata of the *Lhx5* signal are still present (Figure 8q–s). We noted the labeling of the basal preisthmus with *Nts, Pou6f2, Sst*, and *Zfhx3*.

At E18.5, the morphology of the midbrain is nearly mature and many more details of the final structure can be first observed (notably, the development of the PAG). The alar preisthmus thus starts to be differentiable into periaqueductal, intermediate, and superficial strata. A distinct *Trh*-positive basal periaqueductal nucleus was observed. The main novelty was the bulging inwards of the dorsomedial toral region, which evolves from E16.5 onwards, to be discussed in the last section below (T; Figure 9t–w, Figure 10a–h and Figure 11e–y).

At postnatal stages and in topologically horizontal sections (insofar as they show parts of the hindbrain ‘ventrally’, and the actually rostral midbrain ‘dorsally’; see clarificatory schema in [27]), the preisthmus appears at the back of the IC as a concave radial arch that partially embraces the medial and ventral aspects of the IC, stretching from the caudal PAG (part of which is preisthmic) into the classic cuneiform area, which we divide into the intermediate cuneiform nucleus (CnF) and the superficial subbrachial nucleus (SubB) (Figure 12f–i). It limits ‘ventrally’ (in fact caudally) with the isthmic region, where the parabigeminal and microcellular tegmental nuclei plus the isthmic reticular formation can be identified as superficial and intermediate entities. The IC, divided into core and shell (cortical) formations, sits directly on top of (rostral to) the CnF and SubB, and displays its own PAG sector above the preisthmic PAG (Figure 12f). The basal part of the preisthmus is a narrow transverse retrorubral band intercalated between the oculomotor (m1) and trochlear (rh0) nerve nuclei, and between the dorsal and ventral tegmental decussations (in m1) and the decussation of the brachium conjunctivum (rh0; Figs,12a,b). The basal domain is slightly more populated and less narrow at the periventricular PAG level than elsewhere. The positive alar preisthmus markers we illustrated are *Dach1, Ebf3, En1, Gfra1, Id4, Kctd4, Nts, Otx2, Penk, Pitx1, Rgs4, Six3*, and *Slc17a8* (some of these are found only in the alar preisthmic PAG). *Cbln2, Irx1*, and *Tcf7l2* were shown as negative markers at these stages. The basal preisthmus was identified with *Calb1, Cck*, and *Sst*. Note [21] and [27] showed additional dopaminergic and serotonergic markers.

On the whole, the anatomic assessment of the preisthmus first reported by Hidalgo-Sánchez et al. [15] in the chick and Puelles et al. [21] in the mouse was corroborated. The illustrated markers of this territory can help in experimental embryological studies of this territory and may help as well with appropriate connectivity and functional studies on preisthmic populations. We think that the novel findings relative to the existence of distinct topologically AP sectors of the PAG corresponding to the preisthmus and the IC are of interest for the workers in this field.

### 4.5. The Issue of the Dorsomedial Toral Bulges

This is an issue that had not been detected before. While examining E13.5 specimens, we wondered about the meaning and fate of the undifferentiated area dorsomedial to the preisthmus, which reaches the roof plate. This area expressed *Otx2, En1,* and *Pax5* and resembled the preisthmic profile (Figure 4c,m and Figure 5a,b). At E15.5, the aspect of this area was similar, as shown labeled with Otx2, Tcf7l2, and Lhx2 (e.g., Figure 7m,n and Figure 8i,j,m,n). However, at E18.5, distinct development of the dorsomedial caudalmost area had occurred, producing massive rounded bulges that protruded toroidal-like into the ventricular cavity. Note that adult mammals lack such midbrain ventricular bulges, but their reptilian ancestors probably did have them, as do all present-day amphibians, reptiles, and birds [46]. The present late embryonic toral bulges in the mouse are in a way a reminiscence of ancestral conditions.

Analysis at postnatal stages illustrated the rapid morphogenetic disappearance of the toral bulges over the postnatal P1-P4 period. This morphogenetic effect may be due to stretching produced in the dorsomedial toral area by the massive growth of the inferior colliculus (and the latter being pushed back by the parallel massive caudal growth of the superior colliculus). The region first loses the intraventricular protrusions (Figure 13a–f,j) and then continues thinning progressively. It persists in the adult as a tenuous membrane usually ascribed to the roof plate, which is continuous with the similarly thinned isthmic roof plate (Figure 13m). Classical neuroanatomy called this thin membrane the ‘velum medullare superior’ of the cerebellum, though, as we see, it is properly not a part of the cerebellum.

It may be discussed whether the toral region should be ascribed to the dorsal end of the preisthmus within m2 or whether it represents merely a molecularly peculiar extension of the inferior colliculus within the caudalmost dorsal m1. The early pattern of *Otx2* at E13.5, when the IC domain has already reduced its earlier *Otx2* signal to its ventricular zone (Figure 4a–g), suggests that the *Otx2*-positive toral region is attached to the positive preisthmus. However, the toral area is undifferentiated at this stage, and therefore it might represent an area of the IC that still has not lost its early *Otx2* signal. Possibly for the same reason, the toral expression of *Pax5* and *En1* is continuous with that of the preisthmus and no such signal appears at the incipiently differentiated IC primordium (Figure 4i,l,m and Figure 5a,b), while the selective IC labeling with *Tcf7l2* does not extend initially into the toral area (Figure 5aa,bb). At E13.5 and E14.5, the selective IC markers *Foxp2* and *Tcf7l2* do not extend yet as far caudally as the toral area (see Allen data). However, at E15.5 the *Tcf7l2*-positive IC mantle starts to extend towards the toral area, finally occupying it fully over E16.5 and E17.5, thus creating the pattern shown at E18.5 (Allen data). We thus think that the most reasonable interpretation is that the toral area belongs to the IC within m1, being its caudalmost, dorsalmost, and latest differentiated part. Possibly due to its closeness to the isthmic organizer, it may be subjected at early stages to anti-neurogenic signaling, perhaps vehiculated by the *En1* and *Pax5* genes, if not by *Wnt1*, and perhaps also making it susceptible to subsequent postnatal degeneration.

## 5. Conclusions

Our studies enrich available genoarchitectural knowledge on the developing preisthmic midbrain domain in the mouse and establish on a strong basis its early developmental boundaries with regard to both the isthmic hindbrain and the rest of the midbrain, starting with the rostrally adjacent auditory midbrain and the rubral m1 tegmentum. The possibility that the preisthmus contains the progeny of the m2 mesomere postulated by von Kupffer, Palmgren, and Vaage [17,18,32] is supported, thus falsifying their hypothesis that this neuromere is atrophic and lacks a progeny (as was concluded previously in the chick [15,47]). Despite its reduced overall size, the collective domain formed by alar and basal subdomains of the molecularly defined preisthmus fulfills the completion criterium of Puelles and Rubenstein [43], thus supporting its neuromeric nature (i.e., as a complete AP sector of the neural tube). The molecular distinctness and singular histic fate of the preisthmus is attributed to an early effect of high concentrations of the isthmic organizer morphogens FGF8 and WNT1 within competent midbrain-specified neuroepithelium (Figure 1). Our analysis suggests that the midbrain gets divided minimally into four distinct AP regions. The caudalmost one is represented singly by the preisthmus (m2), while the larger m1 neuromeric unit gets divided into at least three AP regions identified by their alar components as inferior colliculus, superior colliculus, and tectal gray, whose full dorsoventral partition is not yet fully resolved.

## Figures and Tables

**Figure 1 ijms-24-09769-f001:**
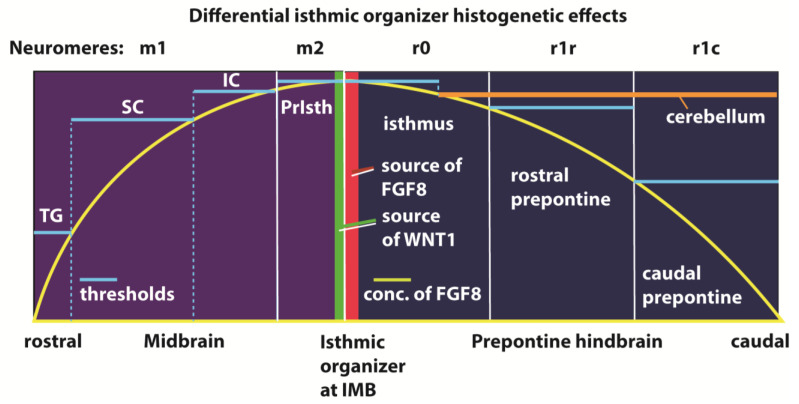
Schematic drawing representing in lateral view (rostral to the left, dorsal above) the midbrain (violet background) and the prepontine hindbrain (dark blue) with identification of anteroposterior areas differentially patterned by the isthmic organizer centered at the isthmo-mesencephalic transverse boundary (IMB). The neuromeric units in this region (mesomeres m1 and m2, and rhombomeres r0, r1r, and r1c) are marked above and appear delimited by transverse white interneuromeric borders. At the organizer, the isthmic source of diffusible FGF8 morphogen is represented by a thick red bar, while the preisthmic source of WNT1 appears as an adjacent green bar. The diffusion curve of FGF8 signal (the yellow line) is represented as decreasing in FGF8 concentration as the distance from the source increases rostral- or caudalwards. Competent neuroepithelial progenitor cells sensitive to this signal are thought to react specifically at particular thresholds of FGF8 concentration (light blue lines), activating different sets of downstream genes within particular ranges of distance from the organizer. These guide differentially local histogenesis into distinct structural and functional fates. Highest levels of FGF8 signal generate the preisthmus (PrIsth in m2) and the isthmic nuclei (within the isthmus; at slightly lower levels of FGF8 the caudal isthmus also participates in the formation of the rostralmost cerebellum; orange bar). Rostral and caudal prepontine nuclei plus other parts of the cerebellum are formed in the rostral and caudal r1 units. Rostrally to the preisthmus, the m1 unit divides its alar plate domain into three structurally different areas: the auditory inferior colliculus (IC), the visual superior colliculus (SC), and the likewise visual tectal gray formation (TG) next to the caudal diencephalon, normally not influenced by the isthmic organizer. The respective limits of these m1 derivatives are traced with blue dash-lines since they are not neuromeric boundaries, but secondary intraneuromeric ones. Apparently m1 having a larger area than m2 allows more subdivisions (more distinct thresholds are possible).

**Figure 2 ijms-24-09769-f002:**
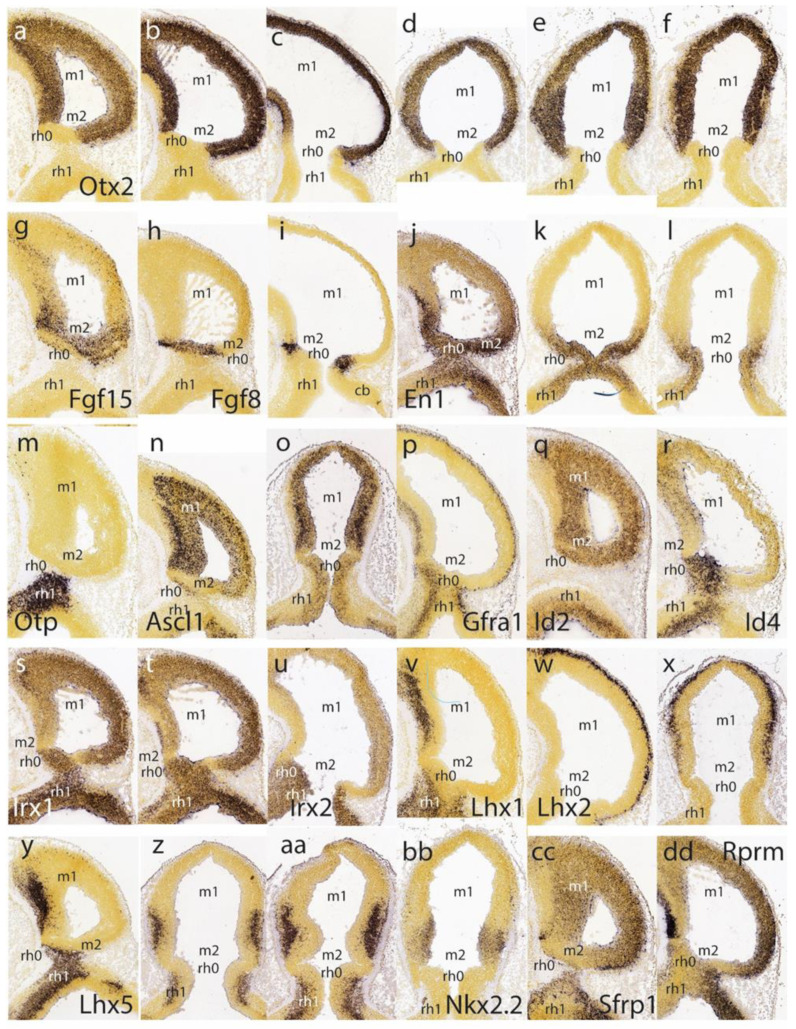
Plate illustrating diverse E11.5 mouse brain sagittal (**a**–**c**,**g**–**j**,**m**,**n**,**p**–**w**,**y**,**cc**,**dd**) or horizontal (**k**,**l**,**o**,**x**,**z**–**bb**) sections reacted for the gene markers indicated at the bottom. The gene name is indicated only in the first image of a series. In sagittal sections rostral is to the left and dorsal upwards, whereas in horizontal sections rostral is up and caudal at the bottom. The magnification is the same in all cases. Note that the horizontal sections clearly show a caudal indentation of the midbrain ventricular surface, coinciding with our ‘m2′ tag; this is the landmark that classic authors denominated ‘sulcus intraencephalicus posterior’ and associated with the postulated m2 neuromere.

**Figure 3 ijms-24-09769-f003:**
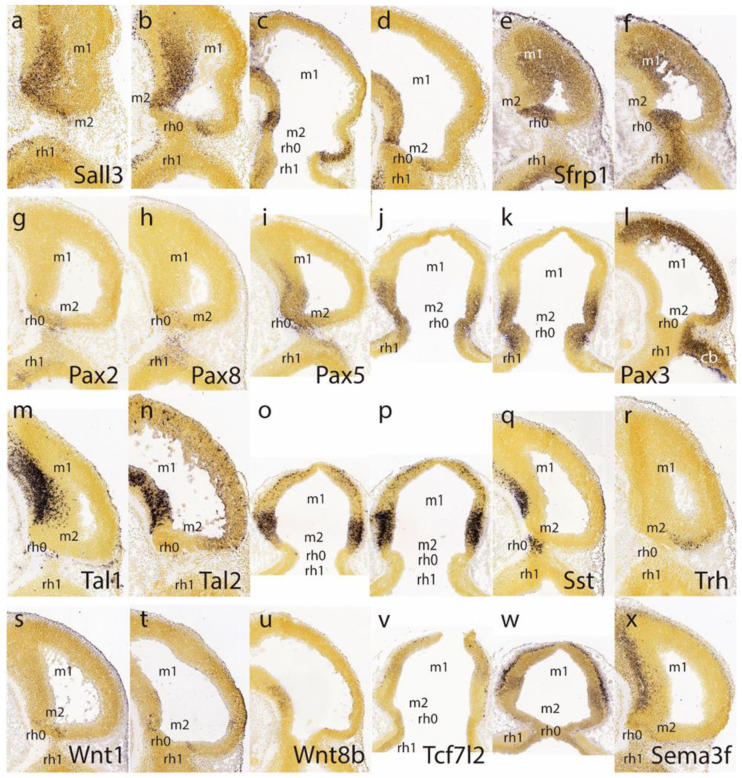
Plate illustrating diverse E11.5 mouse brain sagittal (**a**–**i**,**l**–**n**,**q**–**u**,**x**) or horizontal (**j**,**k**,**o**,**p**,**v**,**w**) sections reacted for the gene markers indicated at the bottom. The gene name is indicated only in the first image of a series. In sagittal sections rostral is to the left and dorsal upwards, whereas in horizontal sections rostral is up and caudal at the bottom. The magnification is the same in all cases.

**Figure 4 ijms-24-09769-f004:**
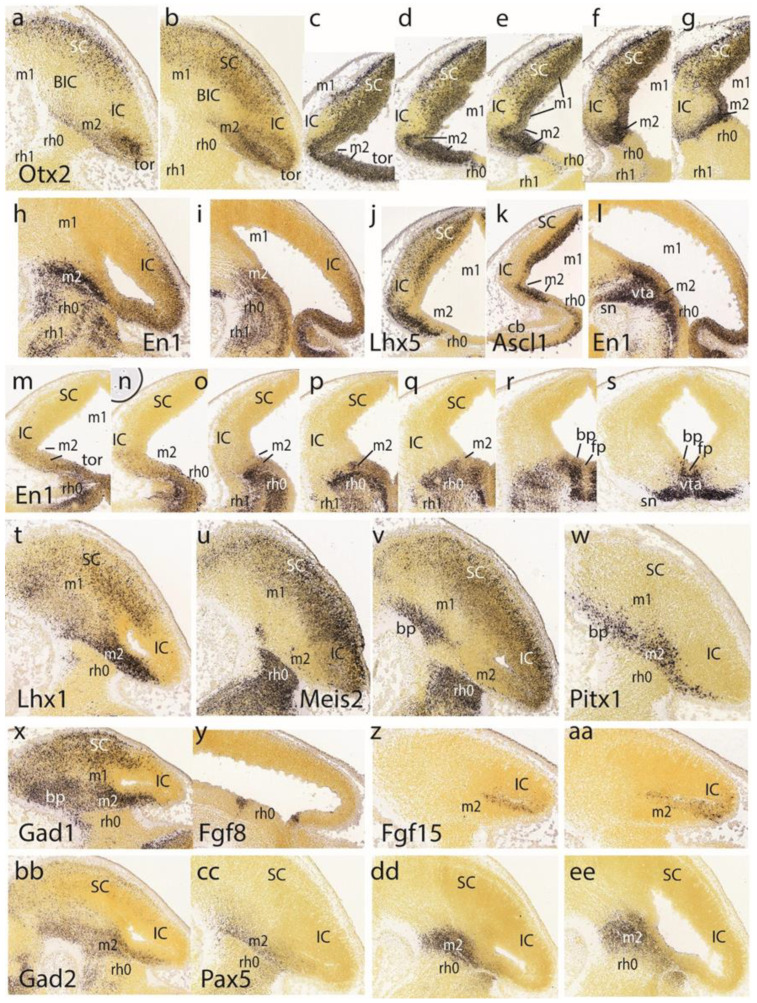
Plate illustrating diverse E13.5 mouse brain sagittal (**a**,**b**,**h**,**i**,**l**,**t**–**ee**) or horizontal (**c**–**g**,**j**,**k**,**m**–**s**) sections reacted for the gene markers indicated at the bottom. The gene name is indicated only in the first image of a series. In sagittal sections rostral is to the left and dorsal upwards, whereas in horizontal sections rostral is up and caudal at the bottom. The magnification is the same in all cases.

**Figure 5 ijms-24-09769-f005:**
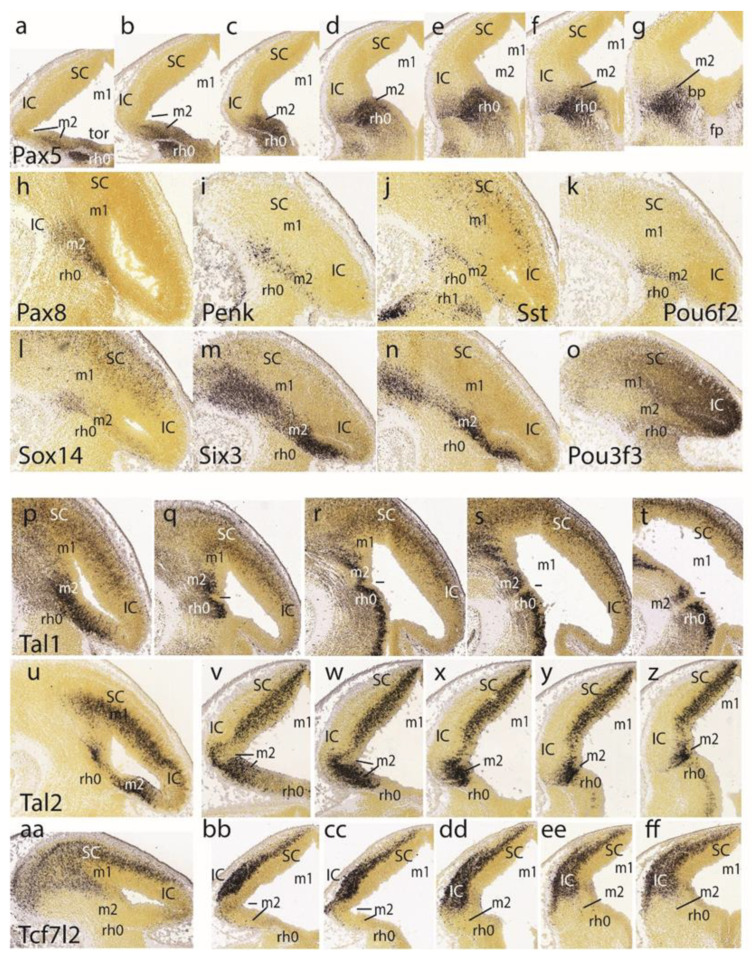
Plate illustrating diverse E13.5 mouse brain sagittal (**h**–**u**,**aa**) or horizontal (**a**–**g**,**v**–**z**,**bb**–**ff**) sections reacted for the gene markers indicated at the bottom. The gene name is indicated only in the first image of a series. In sagittal sections rostral is to the left and dorsal upwards, whereas in horizontal sections rostral is up and caudal at the bottom. The magnification is the same in all cases.

**Figure 6 ijms-24-09769-f006:**
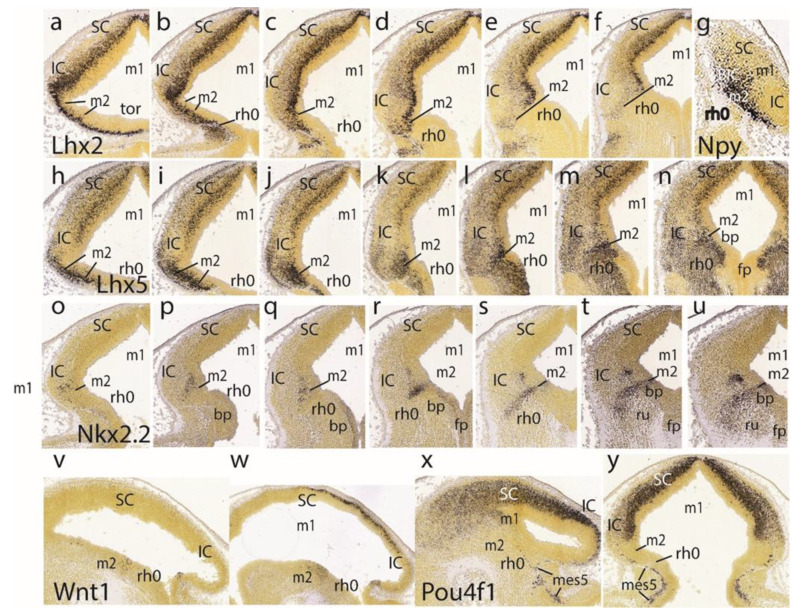
Plate illustrating diverse E13.5 mouse brain sagittal (**g**,**v**–**x**) or horizontal (**a**–**f**,**h**–**u**,**y**) sections reacted for the gene markers indicated at the bottom. The gene name is indicated only in the first image of a series. In sagittal sections rostral is to the left and dorsal upwards, whereas in horizontal sections rostral is up and caudal at the bottom. The magnification is the same in all cases.

**Figure 7 ijms-24-09769-f007:**
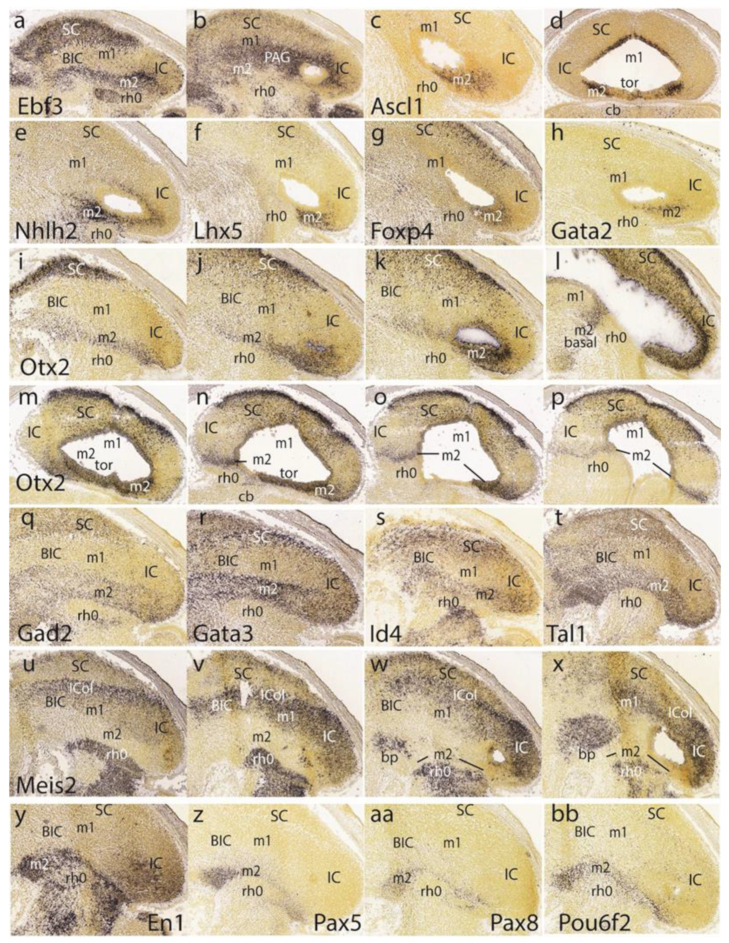
Plate illustrating diverse E15.5 mouse brain sagittal (**a**–**c**,**e**–**l**,**q**–**bb**) or horizontal (**d**,**m**–**p**) sections reacted for the gene markers indicated at the bottom. The gene name is indicated only in the first image of a series. In sagittal sections rostral is to the left and dorsal upwards, whereas in horizontal sections rostral is up and caudal at the bottom. The magnification is the same in all cases.

**Figure 8 ijms-24-09769-f008:**
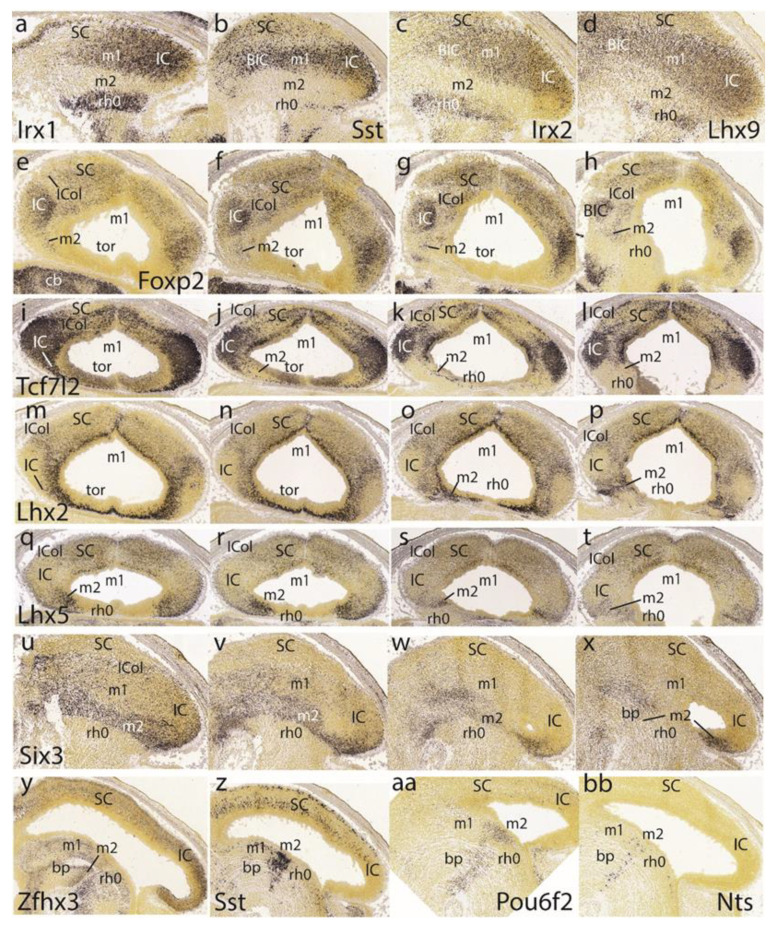
Plate illustrating diverse E15.5 mouse brain sagittal (**a**–**d**,**u**–**bb**) or horizontal (**e**–**t**) sections reacted for the gene markers indicated at the bottom. The gene name is indicated only in the first image of a series. In sagittal sections rostral is to the left and dorsal upwards, whereas in horizontal sections rostral is up and caudal at the bottom. The magnification is the same in all cases.

**Figure 9 ijms-24-09769-f009:**
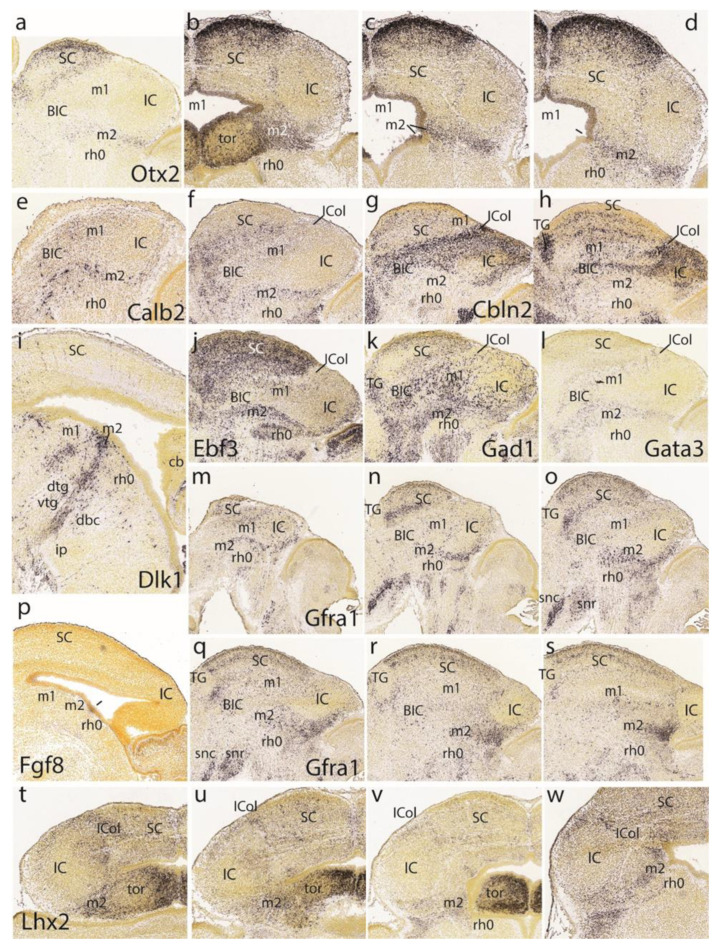
Plate illustrating diverse E18.5 mouse brain sagittal (**a**,**e**–**s**) or horizontal (**t**–**w**) sections reacted for the gene markers indicated at the bottom. The gene name is indicated only in the first image of a series. In sagittal sections rostral is to the left and dorsal upwards, whereas in horizontal sections rostral is up and caudal at the bottom. The magnification is the same in all cases.

**Figure 10 ijms-24-09769-f010:**
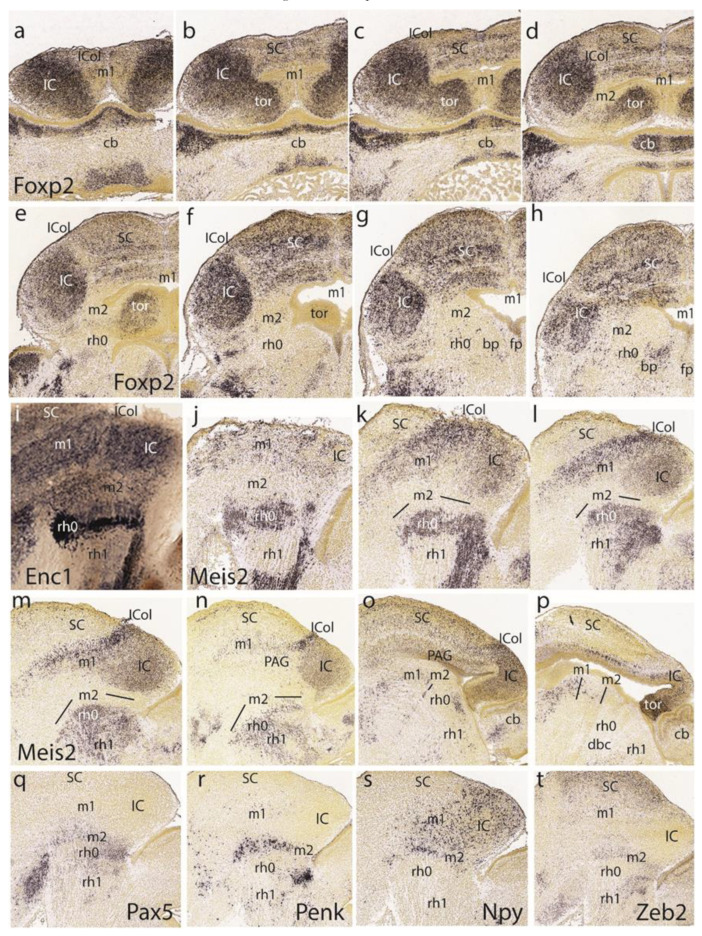
Plate illustrating diverse E18.5 mouse brain sagittal (**i**–**t**) or horizontal (**a**–**h**) sections reacted for the gene markers indicated at the bottom. The gene name is indicated only in the first image of a series. In sagittal sections rostral is to the left and dorsal upwards, whereas in horizontal sections rostral is up and caudal at the bottom. The magnification is the same in all cases.

**Figure 11 ijms-24-09769-f011:**
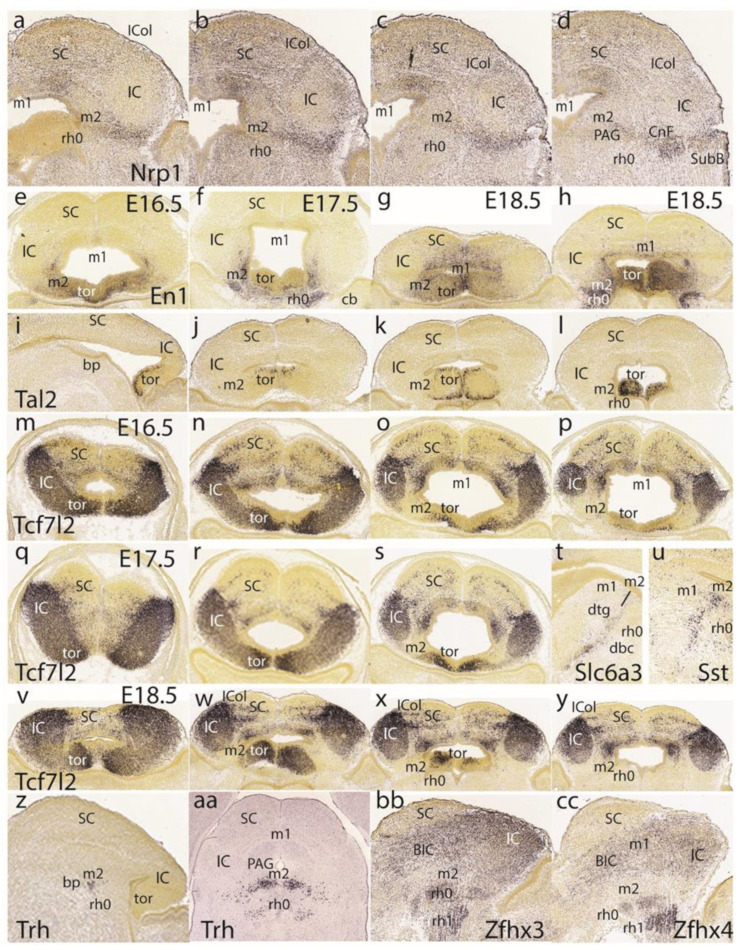
Plate illustrating diverse E18.5 mouse brain sagittal (**i**,**t**,**u**,**z**,**bb**,**cc**) or horizontal (**a**–**h**,**j**–**s**,**v**–**y**,**aa**) sections reacted for the gene markers indicated at the bottom. The gene name is indicated only in the first image of a series. In sagittal sections rostral is to the left and dorsal upwards, whereas in horizontal sections rostral is up and caudal at the bottom. The magnification is the same in all cases.

**Figure 12 ijms-24-09769-f012:**
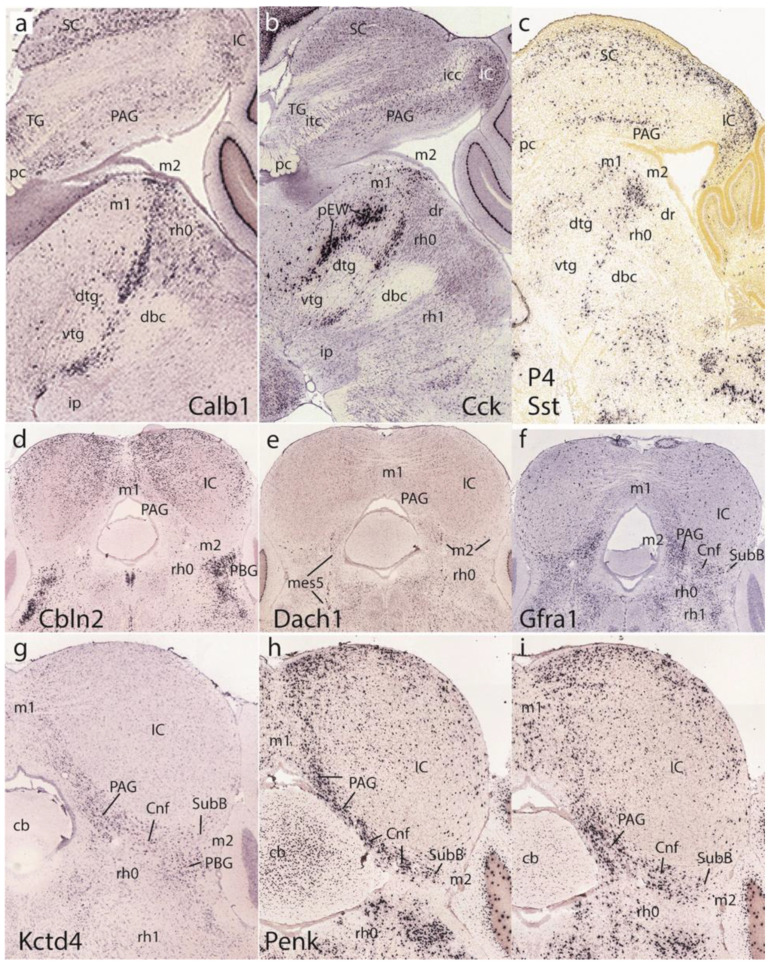
Plate illustrating diverse postnatal mouse brain sagittal (**a**–**c**) or horizontal (**d**–**i**) sections reacted for the gene markers indicated at the bottom. The gene name is indicated only in the first image of a series. In sagittal sections rostral is to the left and dorsal upwards, whereas in horizontal sections rostral is up and caudal at the bottom. The magnification is the same in all cases.

**Figure 13 ijms-24-09769-f013:**
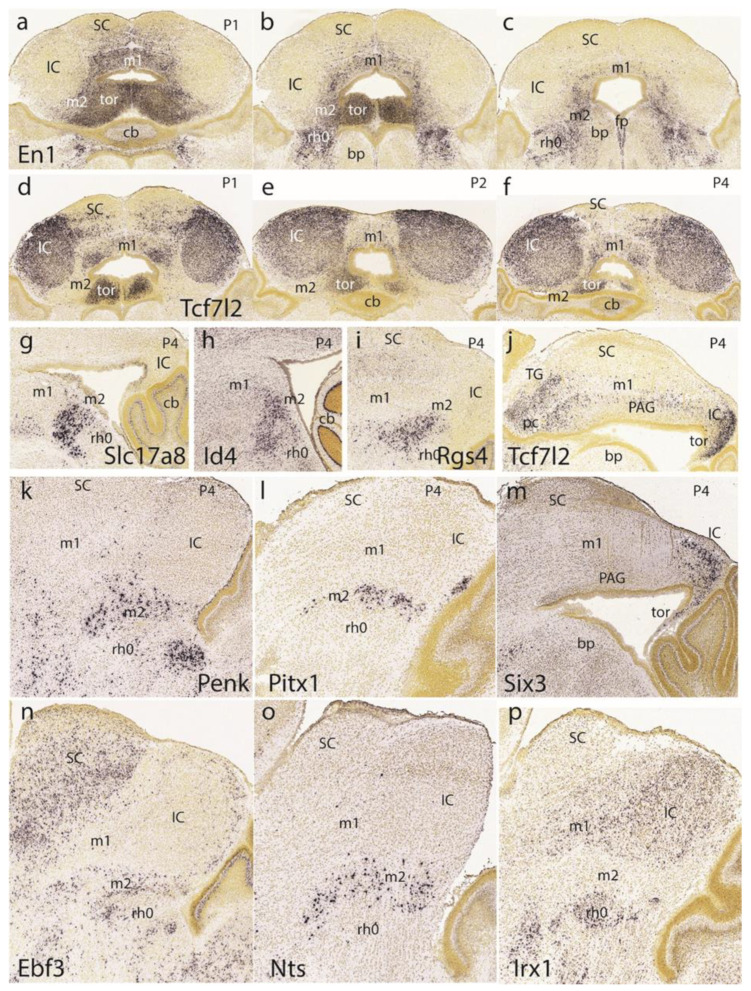
Plate illustrating diverse postnatal mouse brain sagittal (**g**–**p**) or horizontal (**a**–**f**) sections reacted for the gene markers indicated at the bottom. The gene name is indicated only in the first image of a series. In sagittal sections rostral is to the left and dorsal upwards, whereas in horizontal sections rostral is up and caudal at the bottom. The magnification is the same in all cases.

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
