# Peer review of "The Midbrain Preisthmus: A Poorly Known Effect of the Isthmic Organizer"

_ijms, 2023, doi:10.3390/ijms24119769_

Round 1

Reviewer 1 Report

In the manuscript by Puelles et al., the authors provide genoarchitecture evidence of the presence of preisthmus present in the caudal midbrain in vertebrates. The authors used the Allen Developing and Adult Mouse Brain Atlas as well as in situ hybridization of several markers from their own studies during mid-late gestation and post-natal stages of mouse development to illustrate the molecular signature of preisthmus in the context of surrounding brain regions. This study is a great resource for the scientific community not only interested in preisthmus biology but also to understand the expression pattern of many genes during brain development. The authors have done extensive data mining and characterization to expand our knowledge of the development, origin, and location of the mouse preisthmus. We have some minor suggestions that would help the readers follow and understand this detailed study.

1.     Can the authors make a schematic of the description in the introduction that would make it easy to understand what isthmic organizer and preisthmus are in the brain and where are they located with respect to forebrain, midbrain, and hindbrain? The authors can also consider having multiple schematics to explain the major points in the introduction that could also aid the reader not familiar with the brain structure in interpreting the results.

2.     Can the authors include a schematic of a complete brain section and an inset to orient the reader on the brain images in all figures?

3.     Can the authors highlight specific regions in the brain sections that are referenced in the text throughout the manuscript? For instance, in figure 1n,o, can the authors point to the 'sulcus intraencephalicus posterior'?

4.     The figure numbers are not labelled at the top of the figure.

Author Response

As regards the comments of the first reviewer, we developed a new Figure 1 with schematic content explaining the positions of all the brain areas mentioned in the manuscript and the functioning of the isthmic organizar and its gradientally diffused FGF8 signal, to obtain differential histogenetic patterns at different distances, particularly in the midbrain. This schema also features the preisthmic midbrain área at close distance from the organizar .

As regards the demand to identify the ‘sulcus intraencephalicus posterior’, we included a new sentence in the legend of the Fig.2 (former Fig.1), stating that the diverse horizontal sections contained in this plate all clearly show the cited sulcus, precisely at the level were we inserted the tag ‘m2’. Since it is the only ventricular sulcus visible in the whole midbrain it is easily identifiable in horizontal sections (and it corresponds, as stated in the text, with the postulated concavity of the m2 neuromere). It is more difficult to detect in sagital sections, due to its obliquity.

We did not follow the reviewer’s request to number visibly each plate, since each of them is accompanied by the corresponding legend, where its number is given. Moreover, the instructions for authors did not seem to require that such a numbering be done. All other recommendations from Rev,1 were followed. The List of References was revised and corrected according to the format demanded by the journal. We added a pair of missing References, and eliminated some unnecessary ones.

Following a suggestion by the editor of the Special issue, Ms.Mo, we placed the Table 1 as a Supplementary material.

Reviewer 2 Report

The present manuscript presents a comprehensive descriptive analysis of a large number of gene markers to characterize the “preisthmus” domain at the caudal region of  the midbrain in the embryonic mouse. The authors propose the distinction of four regions along the rostro-caudal axis of the mid-brain, with the preisthmus  representing the last one (most caudal). This is a remarkable finding of interest in developmental neurobiology. The manuscript is clearly written manuscript and contains excellent illustrations. I support its publication requiring only few minor changes:

-This article would greatly benefit from the addition of summary figures showing the most relevant results. 

-Pg. 10, line 3. Include the reference (Hidalgo-Sánchez et al., 2005) after “in the chick”. 

Pg. 10, lines 5. change “part of the midbrain, analogously as observed above for En1 (note …)” to “part of the midbrain (Figs.2i-k), analogously as observed above for En1 (Figs.1j-l”; note …).  

Pg. 10. Third paragraph. The expression pattern of Wnt8b is not described (Fig.2u). 

The list of references should be revised. 

Author Response

We accepted and introduced in the revised text all corrections suggested by Reviewer 2 (only some Figure numbers needed to be changed due to the introduction of a new Fig.1 requested by Reviewer 1)

The List of References (alluded to by Reviewer 2) was revised and corrected according to the format demanded by the journal. We added a pair of missing References, and eliminated some unnecessary ones.

Following a suggestion by the editor of the Special issue, Ms.Mo, we placed the Table 1 as a Supplementary material.